# Beyond Distribution Shift: Spurious Features Through the Lens of Training Dynamics

**Nihal Murali**                                                    *nihal.murali@pitt.edu*
*Intelligent Systems Program*
*University of Pittsburgh*

**Aahlad Puli**                                                         *aahlad@nyu.edu*
*Department of Computer Science*
*New York University*

**Ke Yu**                                                                 *yu.ke@pitt.edu*
*Intelligent Systems Program*
*University of Pittsburgh*

**Rajesh Ranganath**                                          *rajeshr@cims.nyu.edu*
*Department of Computer Science*
*New York University*

**Kayhan Batmanghelich**                                        *batman@bu.edu*
*Department of Electrical and Computer Engineering*
*Boston University*

**Reviewed on OpenReview:** *https: // openreview. net/ forum? id= Tkvmt9nDmB*

## Abstract

Deep Neural Networks (DNNs) are prone to learning spurious features that correlate with the label during training but are irrelevant to the learning problem. This hurts model generalization and poses problems when deploying them in safety-critical applications. This paper aims to better understand the effects of spurious features through the lens of the learning dynamics of the internal neurons during the training process. We make the following observations: (1) While previous works highlight the harmful effects of spurious features on the generalization ability of DNNs, we emphasize that not all spurious features are harmful. Spurious features can be "*benign*" or "*harmful*" depending on whether they are "harder" or "easier" to learn than the core features for a given model. This definition is model and dataset dependent. (2) We build upon this premise and use *instance difficulty* methods (like Prediction Depth (Baldock et al., 2021)) to quantify "easiness" for a given model and to identify this behavior during the training phase. (3) We empirically show that the harmful spurious features can be detected by observing the learning dynamics of the DNN's *early layers*. In other words, easy features learned by the initial layers of a DNN early during the training can (potentially) hurt model generalization. We verify our claims on medical and vision datasets, both simulated and real, and justify the empirical success of our hypothesis by showing the theoretical connections between Prediction Depth and information-theoretic concepts like $\mathcal{V}$-usable information (Ethayarajh et al., 2021). Lastly, our experiments show that monitoring only accuracy during training (as is common in machine learning pipelines) is insufficient to detect spurious features. We, therefore, highlight the need for monitoring early training dynamics using suitable instance difficulty metrics.

## 1 Introduction

DNNs tend to rely on spurious features even in the presence of *core* features that generalize well, which poses serious problems when deploying them in safety-critical applications such as finance, healthcare, and autonomous driving (Geirhos et al., 2020; Oakden-Rayner et al., 2020; DeGrave et al., 2021). A feature is termed spurious if it is correlated with the label during training but is irrelevant to the learning problem (Saab et al., 2022; Izmailov et al., 2022). Previous works use a distribution shift approach to explain this phenomenon (Kirichenko et al., 2022; Wiles et al., 2021; Bellamy et al., 2022; Adnan et al., 2022). However, we get additional insights by emphasizing on the learnability and difficulty of these features. We find that not all spurious features are harmful. Spurious features can be "*benign*" or "*harmful*" depending upon their difficulty with respect to signals that generalize well. We show how monitoring example difficulty metrics like Prediction Depth (PD) (Baldock et al., 2021) can reveal the harmful spurious features quite early during training. Early detection of such features is important as it can help develop intervention schemes to fix the problem early. To the best of our knowledge, we are the first to use the training dynamics of the model to detect spurious feature learning.

The premises that support our hypothesis are as follows: *(P1)* Spurious features hurt generalization only when they are *"easier"* to learn than the core features (see Fig-1). *(P2)* Initial layers of a DNN tend to learn easy features, whereas the later layers tend to learn the harder ones (Zeiler & Fergus, 2014; Baldock et al., 2021). *(P3)* Easy features are learned much earlier than the harder ones during training (Mangalam & Prabhu, 2019; Rahaman et al., 2019). Premises *(P1-3)* lead us to conjecture that: *"Monitoring the easy features learned by the initial layers of a DNN early during the training can help identify the harmful spurious features."*

We make the following observations. First, we show that spurious features can be benign (harmful) depending upon whether they are more challenging (easier) to learn than the core features. Second, we empirically show that our hypothesis works well on medical and vision datasets (sections-4.2,A.3), both simulated and real, regardless of the DNN architecture used. We justify this empirical success by theoretically connecting prediction depth with information-theoretic concepts like $\mathcal{V}$-usable information (Ethayarajh et al., 2021) (sections-3,A.1). Lastly, our experiments highlight that monitoring only accuracy during training, as is common in machine learning pipelines, is insufficient to detect spurious features. In addition, we need to monitor the learning dynamics of the model using suitable instance difficulty metrics to detect harmful spurious features (section-4.3). This will not only save time and computational costs, but also help develop reliable models that do not rely on spurious features.

## 2 Related Work

**Not all spurious features hurt generalization:** Geirhos et al. (2020) define spurious features as features that exist in standard benchmarks but fail to hold in more challenging test conditions. Wiles et al. (2021) view spurious feature learning as a distribution shift problem where two or more attributes are correlated during training but are independent in the test data. Bellamy et al. (2022) use causal diagrams to explain spurious correlations as features that are correlated with the label during training but not during deployment. All these papers characterize spurious correlations purely as a consequence of distribution shift; methods exist to build models robust to such shifts (Arjovsky et al., 2019; Krueger et al., 2021; Puli et al., 2022). The distribution shift viewpoint cannot distinguish between benign and harmful spurious features. In contrast, we stress the learnability and difficulty of spurious features, which helps separate the benign spurious features from the harmful ones (see Fig-1). Previous works like Shah et al. (2020); Scimeca et al. (2021) hint at this by saying that DNNs are biased towards simple solutions, and Dagaev et al. (2021) use the "too-good-to-be-true" prior to emphasize that simple solutions are unlikely to be valid across contexts. Veitch et al. (2021) distinguish various model features using tools from causality and stress test the models for counterfactual invariance. Other works in natural language inference, visual question answering, and action recognition, also assume that simple solutions could be harmful in nature (Sanh et al., 2020; Li & Vasconcelos, 2019; Clark et al., 2019; Cadene et al., 2019; He et al., 2019). We take this line of thought further by hypothesizing that simple solutions or, more explicitly, easy features, which affect the early training dynamics of the model can be harmful in nature. We suggest using suitable example difficulty metrics to measure this effect.

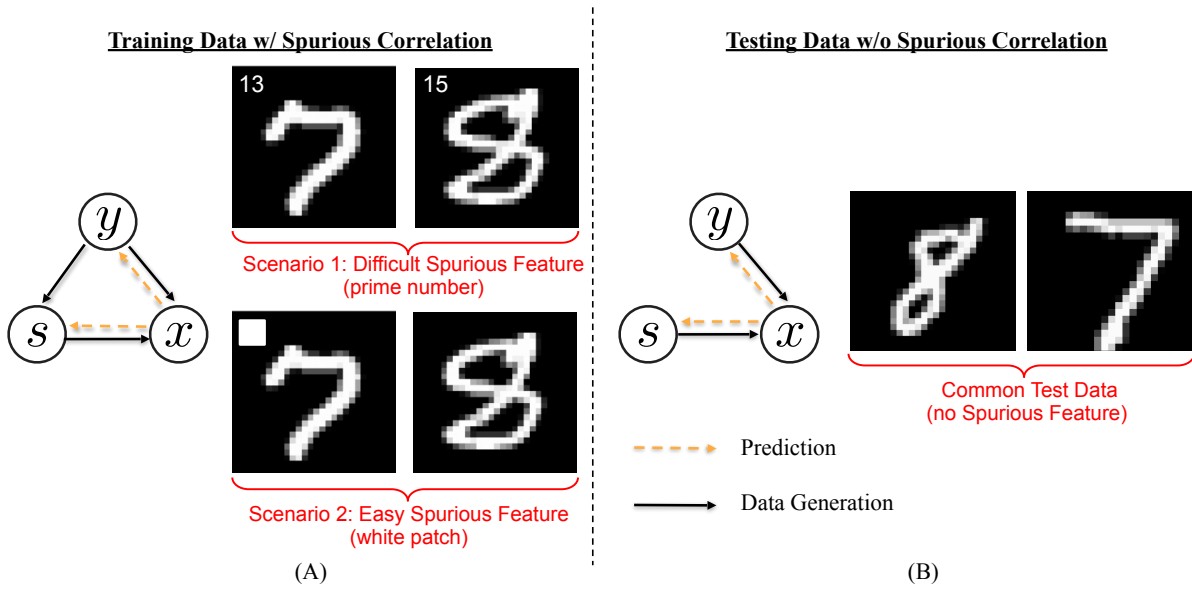

Figure 1: An illustration of how the distribution shift viewpoint cannot distinguish between the benign and the harmful spurious features. This viewpoint suggests that training and testing are different graphical models between input ($x$), output ($y$), and the spurious feature ($s$). If $x$ can predict $s$, and $y$ is not correlated to s on the test data, then $s$ is viewed as a spurious feature. (A) The figure shows two scenarios for even-odd classification. Scenario 1 shows a dataset where all even numbers have a spurious composite number (located at the top-left), and odd numbers have a prime number. Scenario 2 shows a dataset where all odd numbers have a spurious white patch. The spurious white patch is harmful in nature as it is an easy feature that the model can learn, causing poor performance on the test data. Whereas classifying prime numbers, as shown in scenario 1, is challenging. So the model ignores such benign spurious features that do not affect test data performance. This shows not all spurious features hurt generalization.

**Estimating Example Difficulty:** There are different metrics in the literature for measuring instance-specific difficulty (Agarwal et al., 2022; Hooker et al., 2019; Lalor et al., 2018). Jiang et al. (2020) train many models on data subsets of varying sizes to estimate a consistency score that captures the probability of predicting the true label for a particular example. Toneva et al. (2018) define example difficulty as the minimum number of iterations needed for a particular example to be predicted correctly in all subsequent iterations. Agarwal et al. (2022) propose a VoG (variance-of-gradients) score which captures example difficulty by averaging the pre-softmax activation gradients across training checkpoints and image pixels. Feldman & Zhang (2020) use a statistical viewpoint of measuring example difficulty and develop influence functions to estimate the actual leave-one-out influences for various examples. Ethayarajh et al. (2021) use an information-theoretic approach to propose a metric called pointwise $\mathcal{V}$-usable information (PVI) to compute example difficulty. Baldock et al. (2021) define prediction depth (PD) as the minimum number of layers required by the DNN to classify a given input and use this to compute instance difficulty. In our experiments, we use the PD metric to provide a proof of concept for our hypothesis.

**Monitoring Training Dynamics:** Other works that monitor training dynamics have a significantly different goal than ours. While we monitor training dynamics to detect the harmful spurious features, Rabanser et al. (2022) uses neural network training dynamics for selective classification. They use the disagreement between the ground truth label and the intermediate model predictions to reject examples and obtain a favorable accuracy/coverage trade-off. Feng & Tu (2021) use a statistical physics-based approach to study the training dynamics of stochastic gradient descent (SGD). While they study the effect of mislabeled data on SGD training dynamics, we study the effect of spurious features on the early-time learning dynamics of DNNs. Hu et al. (2020) use the early training dynamics of neural networks to show that a simple linear model can often mimic the learning of a two-layer fully connected neural network. Adnan et al. (2022) have

a similar goal as ours and use mutual information to monitor spurious feature learning. However, computing mutual information is intractable for high-dimensional data, and hence their work is only limited to infinite-width neural networks that offer tractable bounds for this computation. Our work, on the contrary, is more general and holds for different neural network architectures and datasets.

## 3 Background and Methodology

In this section, we formalize the notion of spurious features, task difficulty, and harmful vs benign features. Let $P_{tr}$ and $P_{te}$ be the training and test distributions defined over the random variables $\mathbf{X}$ (input), $\mathbf{y}$ (label), and $\mathbf{s}$ (*latent* spurious feature).

**Spurious Feature (s):** A latent feature $\mathbf{s}$ is called spurious if it is correlated with label $\mathbf{y}$ in the training data but not in the test data. Specifically, the joint probability distributions $P_{tr}$ and $P_{te}$ can be factorized as follows.

$$P_{tr}(\mathbf{X}, \mathbf{y}, \mathbf{s}) = P_{tr}(\mathbf{X}|\mathbf{s}, \mathbf{y})P_{tr}(\mathbf{s}|\mathbf{y})P_{tr}(\mathbf{y})$$

$$P_{te}(\mathbf{X}, \mathbf{y}, \mathbf{s}) = P_{tr}(\mathbf{X}|\mathbf{s}, \mathbf{y})P_{te}(\mathbf{s})P_{tr}(\mathbf{y}).$$

The variable $\mathbf{s}$ appears to be related to $\mathbf{y}$ but is not. This is shown in Fig-1. We also introduce notation for task difficulty. The difficulty of a task depends on the model and data distribution $(\mathbf{X}, \mathbf{y})$.

**Task Difficulty ($\Psi$):** Let $\Psi_{\mathcal{M}}^{P}(\mathbf{X}, \mathbf{y})$ indicate the difficulty of predicting label $\mathbf{y}$ from input $\mathbf{X}$ for a model $\mathcal{M}$, such that $\mathbf{X}, \mathbf{y} \sim P$.

We give examples of two metrics that can be used for measuring $\Psi_{\mathcal{M}}^{P}$.

**Example-1 for $\Psi_{\mathcal{M}}^{P}$ (Prediction Depth):** Baldock et al. (2021) proposed the notion of Prediction Depth (PD) to estimate example difficulty. The PD of input is defined as the minimum number of layers the model requires to classify the input. The lower the PD of input, the easier it is to classify. It is computed by building $k$-NN classifiers on the embedding layers of the model, and the earliest layer after which all subsequent $k$-NN predictions remain the same is the PD of the input. Figure-2 illustrates how we compute prediction depth. PD can be mathematically defined as follows:

$$\text{PD}(x) = \min_{k}\{k | f_{knn}^{k}(x) = f_{knn}^{i}(x); i > k\} \tag{1}$$

$f_{knn}$ is the $k$-NN classifier (see Appendix-A.6 for details), $\phi^{i}$ is the feature embedding for the given input at layer-$i$, and $N$ is the index of the final layer of the model. We also use the notion of undefined PD to work with models that are not fully trained. We treat $k$-NN predictions close to 0.5 (for a binary classification setting) as invalid. Figure-3 illustrates how to read the PD plots used in our experiment.

**Example-2 for $\Psi_{\mathcal{M}}^{P}$ ($\mathcal{V}$-Usable Information):** The Mutual Information between input and output, $I(X; Y)$, is invariant with respect to lossless encryption of the input, i.e., $I(\tau(X); Y) = I(X; Y)$. Such a definition assumes unbounded computation and is counter-intuitive to define task difficulty as heavy encryption of $X$ does not change the task difficulty. The notion of *"Usable Information"* introduced by Xu et al. (2020) assumes bounded computation based on the model family $\mathcal{V}$ under consideration. Usable information is measured under a framework called *predictive $\mathcal{V}$-information* (Xu et al., 2020). Ethayarajh et al. (2021) introduce *pointwise $\mathcal{V}$-information* (PVI) for measuring example difficulty.

$$\text{PVI}(x \to y) = -\log_2 g[\phi](y) + \log_2 g'[x](y), \tag{2}$$
$$\text{s.t.} \quad g, g' \in \mathcal{V}$$

The function $g$ is trained on $(\phi, y)$ input-label pairs, where $\phi$ is a null input that provides no information about the label $y$. $g'$ is trained on $(x, y)$ pairs from the training data. Lower PVI instances are harder for $\mathcal{V}$ and vice-versa. Since the first term in Eq-2 corresponding to $g$ is independent of the input $x$, we only consider the second term having $g'$ in our experiments.

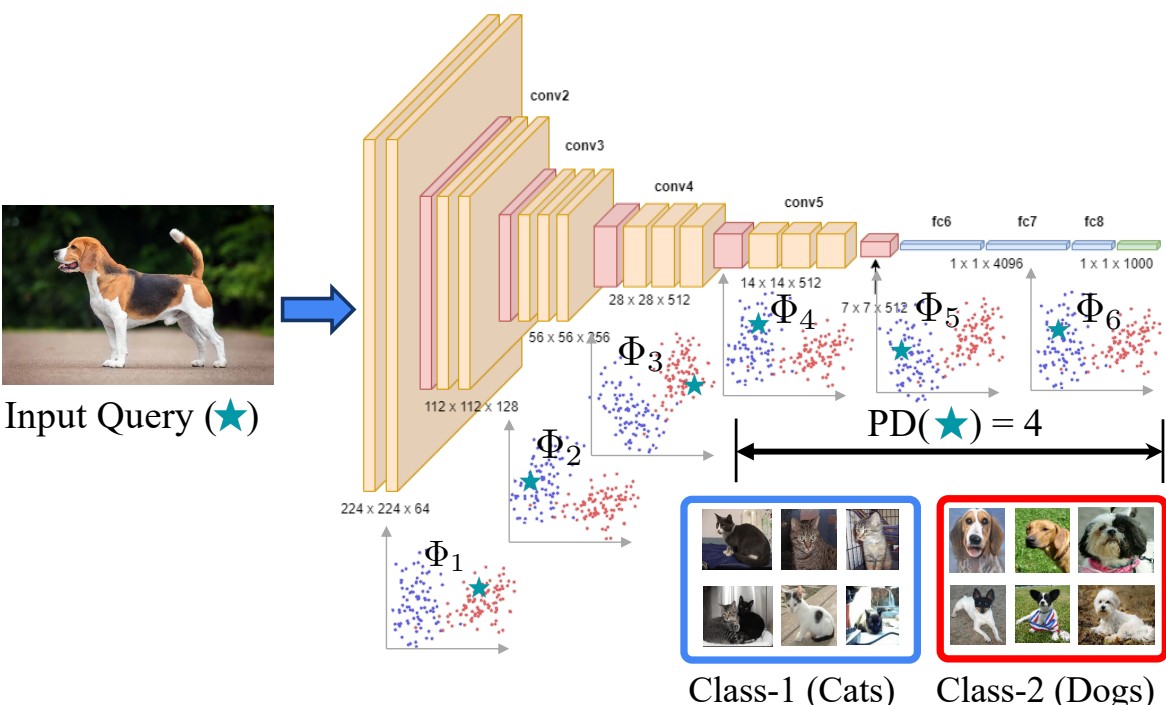

Figure 2: The prediction depth of an image for a given model tells us the minimum number of layers needed by the model to classify the image. A subset of the training data is used to collect feature embeddings at each layer. The KNN classifier uses these feature embeddings at each layer to classify the input query. Prediction depth for the input query is the number of layers after which the KNN predictions converge.

**Harmful Spurious Feature:** The spurious feature $\mathbf{s}$ is harmful for model $\mathcal{M}$ iff $\Psi_{\mathcal{M}}^{P_{tr}}(\mathbf{X}, \mathbf{y}) < \Psi_{\mathcal{M}}^{P_{tr}}(\mathbf{X}, \mathbf{y} | \mathrm{do}(s))$. We use the causal operator "$\mathrm{do}(s)$" to denote a causal intervention on the spurious feature $s$ to break the correlation between spurious features and the labels. This is equivalent to removing the arrow from $\mathbf{y}$ to $s$ as shown in Figure-1B. In other words, if removing the spurious correlation between $\mathbf{s}$ and $\mathbf{y}$ from the training data makes it harder for model $\mathcal{M}$ to predict the true label $\mathbf{y}$, then the spurious feature is "harmful".

The above formalization clarifies that a spurious feature may be harmful to one model but benign for another, as its definition is closely tied to the dataset, model, and task. Figure-4 illustrates this point.

In what follows, we relate the notions of PD and $\mathcal{V}$-usable information. We use $\mathcal{V}_{cnn}$ (of finite depth and width) in our proof as our experiments mainly use CNN architectures.

**Proposition 1:** (Informal) Consider two datasets: $D_s \sim P_{tr}(\mathbf{X}, \mathbf{y})$ with spurious features and $D_i \sim P_{te}(\mathbf{X}, \mathbf{y})$ without them. For some mild assumptions on PD (see Appendix-A.1), if the mean PD of $D_s$ is less than the mean PD of $D_i$, then the $\mathcal{V}_{cnn}$-usable-information for $D_s$ is larger than the $\mathcal{V}_{cnn}$-usable-information for $D_i$: $\mathcal{I}_{\mathcal{V}_{cnn}}^{D_s}(X \to Y) > \mathcal{I}_{\mathcal{V}_{cnn}}^{D_i}(X \to Y)$.

See proof in Appendix-A.1. The proposition intuitively implies that a sufficient gap between the mean PD of spurious and core features can result in spurious features having more $\mathcal{V}_{cnn}$-usable information than core features. In such a scenario, the model will be more inclined to learn spurious features instead of core ones. This proposition justifies using the PD metric to detect spurious features during training, as demonstrated in the following experiments.

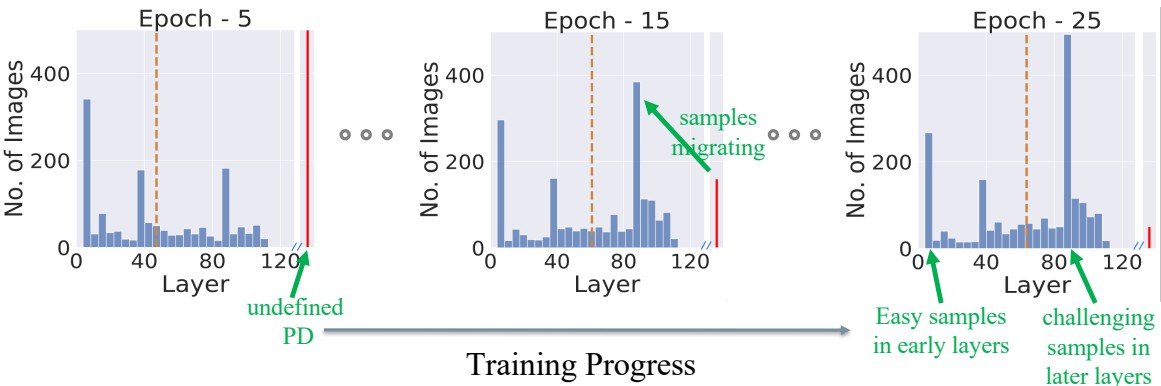

Figure 3: Examples of PD plots (for DenseNet-121) at different stages of the training process. The red bar indicates samples with undefined PD, and the dotted vertical line indicates the mean PD of the plot. Notice that the undefined samples (shown in red) slowly accumulate in layer 88 as training progresses. This is because the model needs more time to learn the challenging samples which accumulate at higher prediction depth, i.e., later layers.

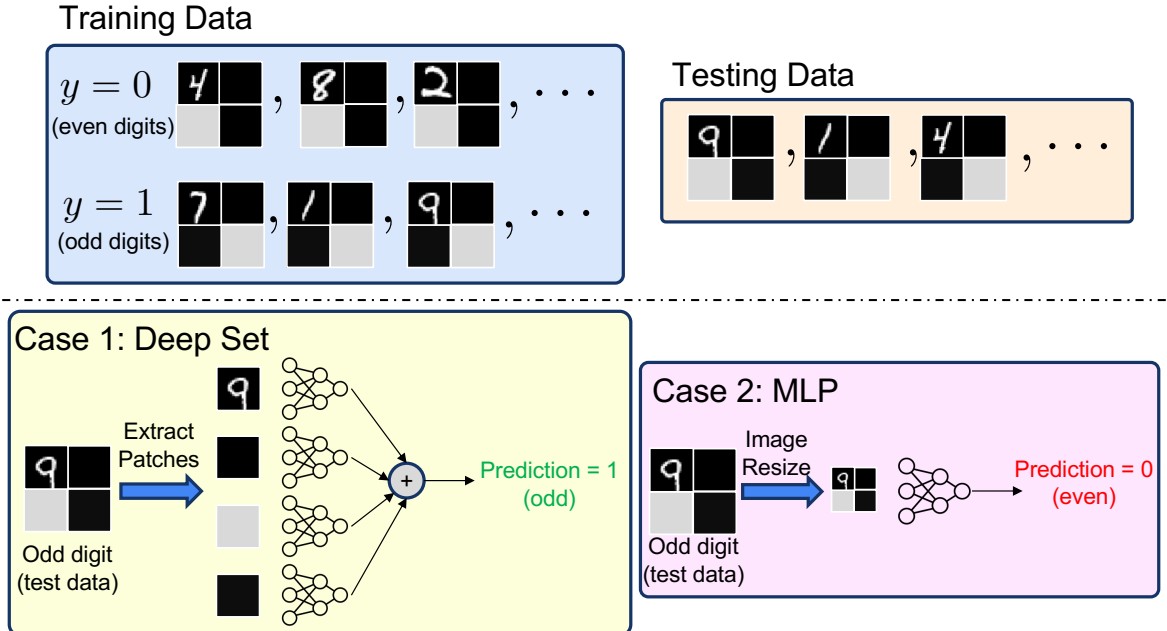

Figure 4: One model's food is another model's poison! A spurious feature may be benign for one model but harmful for another. Consider the task of classifying digits into "even" or "odd" classes. All even (odd) digits in training data have a spurious white patch in the bottom left (right) corner. This correlation doesn't exist in the test data as the spurious white patch is randomly placed. Case-1 shows the Deep Set (Zaheer et al., 2017) model, which patchifies the input image, processes each patch separately, and combines the features. The model's output is invariant to the location of the white patch. However, the MLP model shown in case-2 is location-sensitive; hence, it learns to associate the location of the spurious white patch with the label. Consequently, the MLP model is susceptible to learning such spurious correlations and performs poorly on the test data. So the spurious white patch is *benign* for Deep Sets but *harmful* for the MLP model.

## 4 Experiments

We set up four experiments to evaluate our hypothesis. *First*, we consider two kinds of datasets, one where the spurious feature is easier than the core feature for a ResNet-18 model and another where the spurious feature is harder than the core for the same model. We train a classifier on each dataset and observe that the ResNet-18 model can learn the easy spurious feature but not the harder one. This experiment demonstrates that not all spurious features hurt generalization, but only those spurious features that are easier for the model than the core features are harmful. *Second*, we use medical and vision datasets to demonstrate how monitoring the learning dynamics of the initial layers can reveal harmful spurious features. We show that an early peak in the PD plot may correspond to harmful spurious features and must raise suspicion. Visualization techniques like grad-CAM can provide intuition about what feature is being used at that layer. *Third*, we show how harmful spurious features can often be detected relatively early during training. This is because initial layers which learn such spurious features converge very early during the training. We observe this by monitoring PD plots across training epochs. In all of our experiments, the PD plot reveals the spurious feature within two epochs of training. *Fourth*, we show that datasets with easy spurious features have more "usable information" (Ethayarajh et al., 2021) compared to their counterparts without such features. Due to higher usable information, the model requires fewer layers to classify the images with spurious features. We use this experiment to empirically justify Proposition-1 outlined in Appendix-A.1. We also conduct additional experiments (see Appendix-A.10) to justify the use of the PD metric in our work.

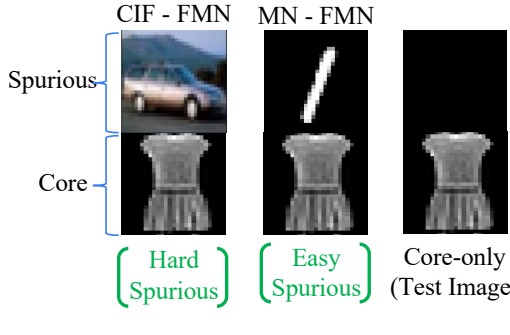

CIF - FMN   MN - FMN

Spurious

Core

Hard Spurious   Easy Spurious   Core-only (Test Image)

Figure 5: Dominoes Dataset

Table 1: Results for the Dominoes experiment averaged across 4-runs. Numbers in bracket show mean-PD (dataset difficulty). Core-only accuracy indicates the model's reliance on core features. Models achieve high core-only accuracy when spurious features are harder than core features.

| Dataset (Spurious-Core) | Is spurious harder than core? | Validation Acc. | Test Acc. (Core-only) |
|---|---|---|---|
| CIF(6.8) - FMN(3.9) | yes | 99.12±0.27% | 98.95±0.30% |
| MN(2.2) - FMN(3.9) | no | 99.95±0.05% | 50.75±2.96% |
| CIF(6.8) - KMN(5) | yes | 98.91±0.16% | 98.30±1.08% |
| MN(2.2) - KMN(5) | no | 99.97±0.05% | 50.48±2.64% |
| CIF(6.8) - MN(2.2) | yes | 99.74±0.07% | 99.5±0.66% |
| KMNpatch(1.1) - MN(2.2) | no | 99.97±0.04% | 68.78±20.03% |

### 4.1 Not all spurious features hurt generalization!

We use the Dominoes binary classification dataset (formed by concatenating two datasets vertically; see Fig-4) similar to the setup of Kirichenko et al. (2022). The bottom (top) image acts as the core (spurious) feature. Images are of size $64 \times 32$. We construct three pairs of domino datasets such that each pair has both a hard and an easy spurious feature with respect to the common core feature (see Table-1). We use classes {0,1} for MNIST and SVHN, {coat,dress} for FMNIST, and {airplane, automobile} for CIFAR10. We also include two classes from Kuzushiji-MNIST (or KMNIST) and construct a modification of this dataset called KMNpatch, which has a spurious patch feature (5x5 white patch on the top-left corner) for one of the two classes of KMNIST. The spurious features are perfectly correlated with the target. The order of dataset difficulty based on the mean-PD is as follows: KMNpatch(1.1) < MNIST(2.2) < FMNIST(3.9) < KMNIST(5) < SVHN(5.9) < CIFAR10(6.8). We use a ResNet18 model and measure the validation and core-only accuracies. The validation accuracy is measured on a held-out dataset sampled from the same distribution. For the core-only (test) accuracy, we blank out the spurious feature (top-half image) by replacing it with zeroes (same as Kirichenko et al. (2022)). The higher the core-only accuracy, the lesser the model's reliance on the spurious feature.

Table 1 shows that all datasets achieve a high validation accuracy as expected. The core-only accuracy stays high (>98%) for datasets where the spurious feature is harder to learn than the core feature, indicating the

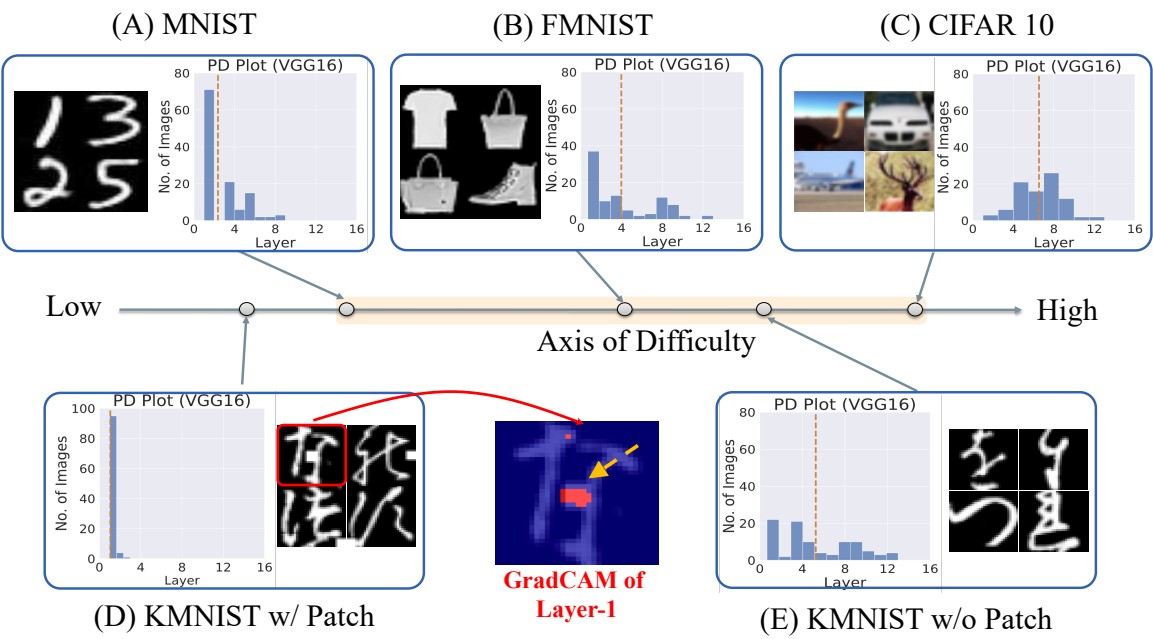

Figure 6: Top row shows three reference datasets and their corresponding prediction depth (PD) plots. The datasets are ordered based on their difficulty (measured using mean PD shown by dotted vertical lines): KMN w/ patch(1.1) < MNIST(2.2) < FMNIST(3.9) < KMN w/o patch(5) < SVHN(5.9) < CIFAR10(6.8). The bottom row shows the effect of the spurious patch on the KMNIST dataset. The yellow region on the axis indicates the expected difficulty of classifying KMNIST. While the original KMNIST lies in the yellow region, the spurious patch significantly reduces the task difficulty. The Grad-CAM shows that the model focuses on the spurious patch.

model's high reliance on core features. When the spurious feature is easier than the core, the model learns to leverage them, and hence the core-only accuracy drops to nearly random chance ($\sim$50%). Interestingly, the KMNpatch-MN results have a much higher core-only accuracy ($\sim$69%) and a more significant standard deviation. We explain this phenomenon in Appendix-A.7. This experiment demonstrates that not all spurious features hurt generalization, and only those that are easier than core features are harmful.

## 4.2 Monitoring Initial Layers Can Reveal the Harmful Spurious Features

**Synthetic Spurious Feature in Toy Dataset:** To provide a proof of concept, we demonstrate our method on the Kuzushiji-MNIST (KMNIST) (Clanuwat et al., 2018) dataset comprising Japanese Hiragana characters. The dataset has ten classes and images of size $28 \times 28$, similar to MNIST. We insert a white patch (spurious feature) at a particular location for each of the ten classes. The location of the patch is class-specific. We train two VGG16 models, one on the KMNIST with a spurious patch ($\mathcal{M}_{sh}$) and another on the original KMNIST without the patch ($\mathcal{M}_{orig}$).

Fig-6 shows the prediction depth (PD) plots for this experiment. The vertical dashed lines show the mean PD for each plot. Fig-6D shows that KMNIST with patch has a much lower mean PD than all the other datasets, and the PD plot is also highly skewed towards the initial layers. This suspicious behavior is because the white patch is a very easy feature, and hence the model only needs a single layer to detect it. The Grad-CAM maps for layer-1 confirm this by showing that $\mathcal{M}_{sh}$ focuses mainly on the patch (see Fig-6D), and hence the test accuracy on the original KMNIST images is very low ($\sim$8%). The PD plot for $\mathcal{M}_{orig}$ (see Fig-6E) is not as skewed toward lower depth as the plot for $\mathcal{M}_{sh}$. This is expected as $\mathcal{M}_{orig}$ is not looking at the spurious patch and therefore utilizes more layers to make the prediction. The mean PD for $\mathcal{M}_{orig}$ suggests that the original KMNIST is harder than Fashion-MNIST but easier than CIFAR10. $\mathcal{M}_{orig}$ also achieves a higher test accuracy ($\sim$98%).

This experiment demonstrates how models that learn spurious features ($\mathcal{M}_{sh}$) exhibit PD plots that are suspiciously skewed towards the initial layers. The skewed PD plot should raise concerns, and visualization techniques like Grad-CAM can aid our intuition about what spurious feature the model may be utilizing at any given layer.

**Semi-Synthetic Spurious Feature in Medical Datasets:** We follow the procedure by DeGrave et al. (2021) to create the ChestX-ray14/GitHub-COVID dataset. This dataset comprises Covid19 positive images from Github Covid repositories and negative images from ChestX-ray14 dataset (Wang et al., 2017b). In addition, we also create the Chex-MIMIC dataset following the procedure by Puli et al. (2022). This dataset comprises 90% images of Pneumonia from Chexpert (Irvin et al., 2019) and 90% healthy images from MIMIC-CXR (Johnson et al., 2019). We train two DenseNet121 models, $\mathcal{M}_{covid}$ on the ChestX-ray14/GitHub-COVID dataset, and $\mathcal{M}_{chex}$ on the Chex-MIMIC dataset. We use DenseNet121, a common and standard architecture for medical image analysis. Images are resized to $512 \times 512$.

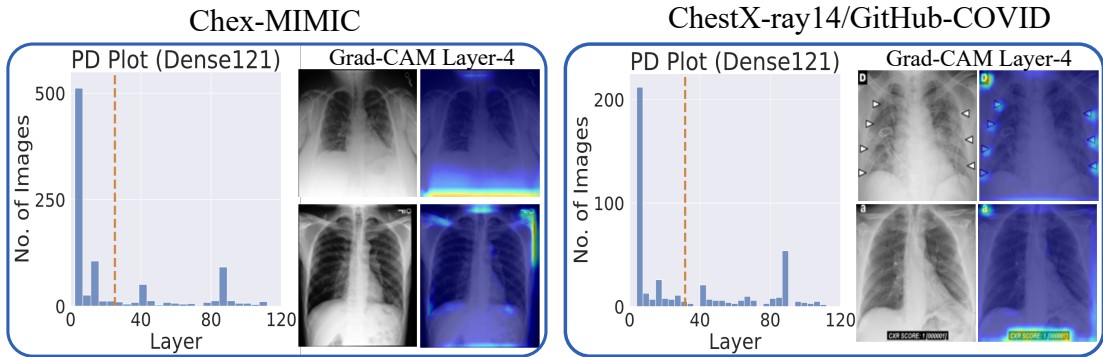

Figure 7: PD plots for two DenseNet-121 models trained on Chex-MIMIC and ChestX-ray14/GitHub-COVID datasets are shown in the figure, along with their corresponding Grad-CAM visualizations. Both PD plots exhibit a very high peak in the initial layers (1 to 4), indicating that the models use very easy features to make the predictions.

Fig-7 shows the PD plots for $\mathcal{M}_{chex}$ and $\mathcal{M}_{covid}$. Both the plots are highly skewed towards initial layers, similar to the KMNIST with spurious patch in Fig-6D. This again indicates that the models are using very easy features to make the predictions, which is concerning as the two tasks (pneumonia and covid19 detection) are hard tasks even for humans. Examining the Grad-CAM maps at layer-4 reveals that these models focus on spurious features outside the lung region, which are irrelevant because both diseases are known to affect mainly the lungs. The reason for this suspicious behavior is that, in both these datasets, the healthy and diseased samples have been acquired from two different sources. This creates a spurious feature because source-specific attributes or tokens are predictive of the disease and can be easily learned, as pointed out by DeGrave et al. (2021).

**Real Spurious Feature in Medical Dataset:** For this experiment, we use the NIH dataset (Wang et al., 2017a) which has the popular chest drain spurious feature (for Pneumothorax detection) (Oakden-Rayner et al., 2020). Chest drains are used to treat positive Pneumothorax cases. Therefore, the presence of a chest drain in the lung is positively correlated with the presence of Pneumothorax and can be used by the deep learning model (Oakden-Rayner et al., 2020). Appendix-A.5 outlines the procedure we use to obtain chest drain annotations for the NIH dataset. We train a DenseNet121 model ($\mathcal{M}_{nih}$) for Pneumothorax detection on NIH images of size $128 \times 128$.

Fig-8A shows the PD plot for $\mathcal{M}_{nih}$. We observe that the distribution is not as skewed as the plots in the previous experiments. This is because all the images come from a single dataset (NIH). However, we do see suspicious peaks at the initial layers. Pneumothorax classification is challenging even for radiologists; hence, peaks at the initial layers raise suspicion. The Grad-CAM maps in Figs-8B & 8C reveal that the initial layers look at irrelevant artifacts and chest drains in the image. This provides evidence that the initial layers are

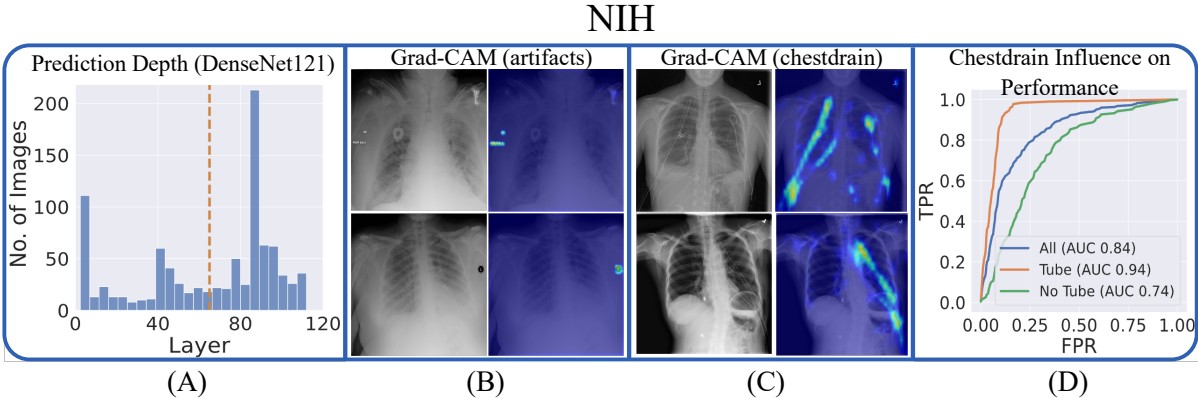

Figure 8: Spurious correlations in NIH dataset. (A) PD plot for DenseNet-121 trained on NIH shows prominent peaks in the initial layers. (B, C) Grad-CAM reveals that the initial layers use irrelevant and spurious artifacts and chest drains for classification. (D) The chest drain spurious feature affects the AUC performance of the model. The X-axis (Y-axis) shows the false positive (true positive) rate.

looking at the harmful spurious features. Fig-8D shows that the AUC performance is 0.94 when the diseased patients have a chest drain and 0.74 when they don't. In both cases, the set of healthy patients remains the same. This observation is consistent with the findings of Oakden-Rayner et al. (2020) and indicates that the model looks at chest drains to classify positive Pneumothorax cases.

The above experiments demonstrate how a peak located in the initial layers of the PD plot should raise suspicion, especially when the classification task is challenging. Visualization techniques like Grad-CAM can further aid our intuition and help identify harmful spurious features being learned by the model. This approach works well even for realistic scenarios and challenging spurious features (like chest drain for Pneumothorax classification), as shown above. Appendix-A.3 includes additional results on vision datasets.

### 4.3 Detecting Harmful Spurious Features Early

Fig-9 shows the evolution of the PD plot across epochs for $\mathcal{M}_{nih}$ (which is the model used in Fig-8). This visualization helps us observe the training dynamics of the various layers. The red bar in the PD plots shows the samples with undefined prediction depths.

These plots reveal several useful insights into the learning dynamics of the model. Firstly, we see three prominent peaks in epoch-1 at layers-4,40,88 (see Fig-9A). The magnitude of the initial peaks (like layers-4&40) remains nearly constant throughout the training. These peaks correspond to spurious features, as discussed in the previous section. This indicates that the harmful spurious features can often be identified early (epoch-1 in this case). Fig-10 shows the PD plots at epoch-2 for other datasets with spurious features. It is clear from Fig-10 that the suspiciously high peak at the initial layer is visible in the *second epoch* itself. The Grad-CAM maps reveal that this layer looks at irrelevant artifacts in the dataset. This behavior is seen in all datasets shown in Fig-10.

Secondly, we also see that accuracy and AUC plots are not sufficient to detect spurious features during training. We need to monitor the training dynamics using suitable metrics (like PD) to detect this behavior. Thirdly, the red peak (undefined samples) decreases in magnitude with time, and we see a proportional increase in the layer-88 peak. This corroborates well with the observation that later layers take more time to converge (Rahaman et al., 2019; Mangalam & Prabhu, 2019; Zeiler & Fergus, 2014). Therefore, samples with higher PD are initially undefined and do not appear in the PD plot. Nonetheless, samples with lower PD show up very early during the training, which helps us detect the harmful spurious features early. Early detection can consequently help develop intervention schemes that fix the problem early.

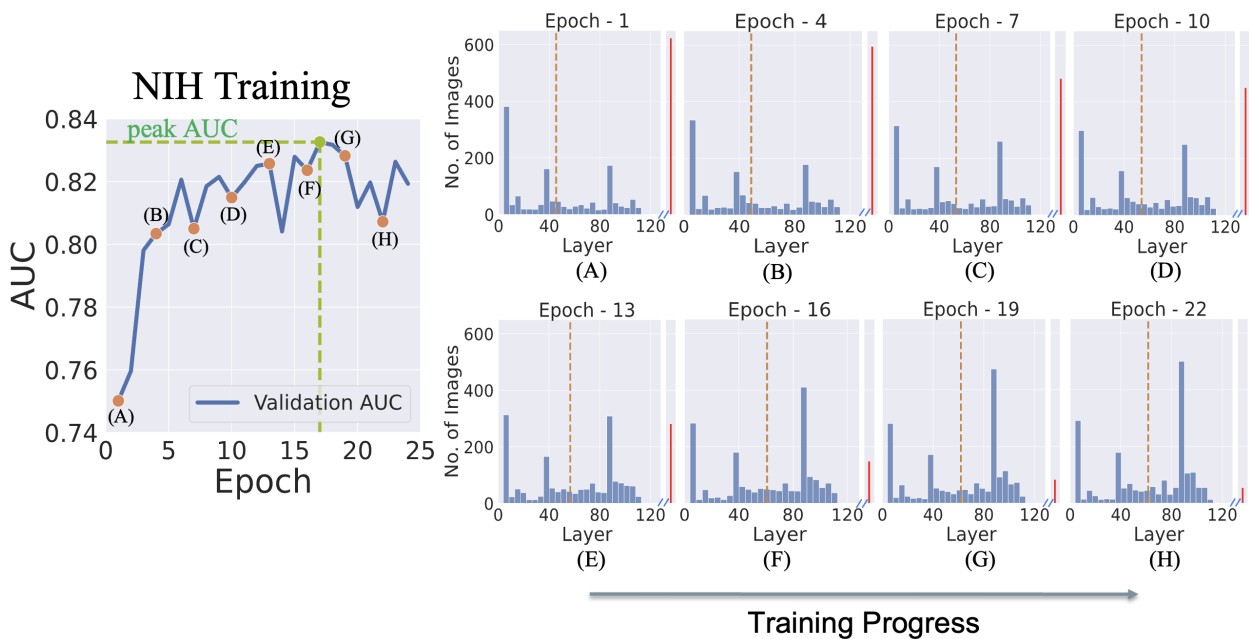

Figure 9: Evolution of PD plot across epochs shows the training dynamics of the DNN on the NIH dataset. The initial peaks (layers-4&40) are relatively stable throughout training, whereas the later peaks (layer-88) change with time. The initial layers learn spurious features, which can be detected early during the training. Samples with undefined PD (shown in red) take more time to converge and eventually accumulate in the later layers (layer 88 in this case).

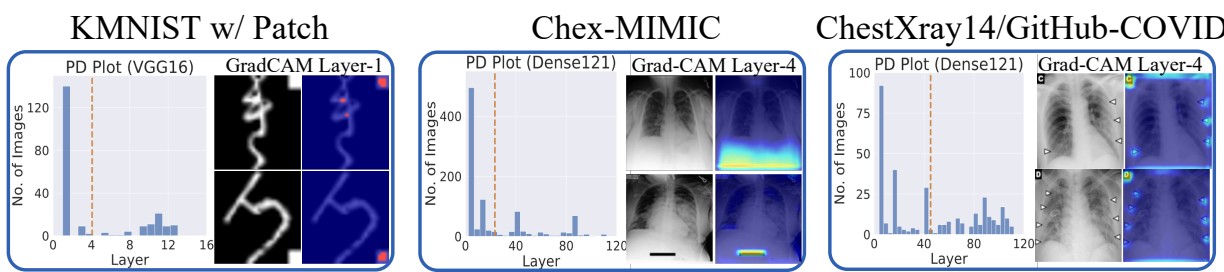

Figure 10: Epoch-2 PD plots for various datasets with spurious features. The high spurious peak in the initial layer is visible in all the datasets indicating that the harmful spurious features can be detected early during the training.

## 4.4 Prediction Depth $\approx \mathcal{V}$-Usable Information

Table-2 measures the influence of spurious features on NIH and KMNIST using PD and PVI metrics. All diseased patients in the "NIH w/ Spurious feat." dataset have a chest drain, whereas all diseased patients in the "NIH w/o Spurious feat." dataset have no chest drain. The set of healthy patients is common for the two datasets. The KMNIST datasets are the same as those used in Section-4.2. We use VGG16 for KMNIST and DenseNet121 for NIH. Other training details are the same as Section-4.2.

Table-2 shows that datasets with spurious features ($D_s$) have smaller mean PD values than their counterparts without such features ($D_i$). Proposition-1 (see Section-3, Appendix-A.1) shows that a sufficient gap between the mean PDs of $D_s$ and $D_i$ causes the $\mathcal{V}$-Information of $D_s$ to be greater than $D_i$. Table-2 confirms this

Table 2: Effect of Spurious features on Prediction Depth and the negative conditional $\mathcal{V}$-entropy ($-H_{\mathcal{V}_{cnn}}(Y \mid X)$). The label marginal distributions are the same with or without the spurious feature, and thus the negative conditional $\mathcal{V}$-entropy is proportional to $\mathcal{V}$-information.

| Dataset | mean PD | $-H_{\mathcal{V}_{cnn}}(Y \mid X)$ |
|---|---|---|
| NIH w/ Spurious feat. | 53.43 | -0.1171 |
| NIH w/o Spurious feat. | 75.33 | -0.2321 |
| KMNIST w/ Spurious feat. | 1.06 | -0.0024 |
| KMNIST w/o Spurious feat. | 5.25 | -0.0585 |

in a medical-imaging dataset with a real chest drain spurious feature, and we see that the mean "usable information" increases when there is a spurious feature. This implies that the model learns spurious features as they have more usable information than the core features. We investigate this relationship between PD and $\mathcal{V}$-information in more detail in Appendix-A.4. Ethayarajh et al. (2021) also show that $\mathcal{V}$-information is positively correlated with test accuracy. This explains the significant change in AUC observed in Fig-8D. Proposition 1 bridges the gap between the notions of PD and $\mathcal{V}$-usable information. This connection between $\mathcal{V}$-information and PD indicates that monitoring early training dynamics using PD not only helps detect the harmful spurious features but also bears insights into the dataset's difficulty (in information-theoretic terms) for a given model class.

## 5 Conclusion

We study spurious features by monitoring the early training dynamics of deep neural networks (DNNs). We empirically show that not all spurious features hurt generalization, but only those that are easier than the core features do. So spurious features can be "benign" or "harmful" depending on whether they are harder or easier than core features for a given model. Our hypothesis is: *"Monitoring the easy features learned by the initial layers of a DNN early during the training can help identify harmful spurious features."* We validate this hypothesis on real medical and vision datasets. We show that spurious features are learned quite early during the training and one can detect them by monitoring the early training of DNNs using instance difficulty metrics like Prediction Depth (PD). Further, we show a theoretical connection between PD and $\mathcal{V}$-information to support our empirical results. Datasets with spurious features have more $\mathcal{V}$-information as compared to their counterparts without such features causing the model to learn the spurious feature. To conclude, relying only on accuracy plots is insufficient for detecting harmful spurious features, and we recommend monitoring the DNN training dynamics using instance difficulty metrics like PD for the early detection of such features.

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

# A  Appendix

## A.1  Proof of Proposition-1

**Proposition A.1.** *Given two datasets, $D_s$ with spurious features and $D_i$ without them, we assume the following:*

1. *(Well-Trained Model Assumption) The part of the network from any representation to the label is one of the functions that compute $\mathcal{V}$-information.*

2. *(Function Class Complexity Assumption) Assume that there exists a $K \in \{1, N\}$ such that $V_{cnn}$ of depth $N - K$ is deep enough to be a strictly larger function class than $V_{knn}$ with a fixed neighbor size (29 in this paper). Assume that this $V_{knn}$ is a larger function class than a linear function.*

3. *(Controlled Confidence Growth Assumption) For both datasets $D \in \{D_s, D_i\}$, assume that the for all $k \in \{1, \cdots, N\}$,*
$$\tau \leq \mathcal{I}^D_{\mathcal{V}_{knn}}(\phi_k) - \mathcal{I}^D_{\mathcal{V}_{knn}}(\phi_{k-1}) \leq \epsilon$$

4. *(Prediction Depth Separation Assumption) Let $L$ be an integer such that, $L \leq K$ and $L < N - \psi \max_y (-\log p(Y = y))$. Note that $p(Y = y)$ is simply the prevalence of class $y$. Let there exist a gap in prediction depths of samples in $D_s$ and $D_i$: $\psi \in (0, 0.5)$ such that $1 - \psi$ fraction of $D_s$ has prediction depth $\leq L$ and $1 - \psi$ fraction of $D_i$ has prediction depth $> K$.*

*Then, for a model class of $N$-layer CNNs, we show that the $\mathcal{V}_{cnn}$-information for $D_s$ is greater than $\mathcal{V}_{cnn}$-information for $D_i$:*
$$\mathcal{I}^{D_s}_{\mathcal{V}_{cnn}}(X \to Y) \geq \mathcal{I}^{D_i}_{\mathcal{V}_{cnn}}(X \to Y)$$

*Proof.* Before proceeding to the proof, we attempt to justify and reason about the above assumptions.

Assumption-1 states a property of trained neural networks in the context of usable information. Let $f(X)$ be a trained neural network. Consider splitting the network into the representation $\phi_k(X)$ at the $k^{th}$ layer and the rest of the network as a function applied to $\phi_k(X)$: i.e., $f(X) = f_k \circ \phi_k(X)$. Then we assume that $f_k(\cdot)$ is the function that achieves the $\mathcal{V}_{\text{cnn of size(n-k)}}$-information between $\phi_k(X)$ and $Y$ (Ethayarajh et al., 2021; Xu et al., 2020). The function that computes $\mathcal{V}$-information must achieve a minimum cross-entropy (Ethayarajh et al., 2021). So if we train $f(X)$ by minimizing the cross-entropy loss, $f_k(.)$ must converge to a function that achieves the $\mathcal{V}_{\text{cnn of size(n-k)}}$-information between $\phi_k(X)$ and $Y$.

Assumption-2 implies that the CNN class in $\mathcal{V}_{cnn}$ is deep enough such that the network after the $K^{th}$ layer can approximate a $k$-NN classifier with 29 neighbors; ($K$ here is same as the $K$ in assumption-4). This is also a reasonable assumption (Chaudhuri & Dasgupta, 2014; Gühring et al., 2020). Chaudhuri & Dasgupta (2014) lower bounds the error for $k$-NN classifiers for a fixed $k$, and Gühring et al. (2020) shows the depth expressivity of CNN classifiers. Assumption-3 states that the difference in $\mathcal{V}_{knn}$-information between intermediate layers does not explode indefinitely and thus can be bounded by some positive quantities $\tau$ and $\epsilon$.

Assumption-4 is also easily satisfied. For example, if the smallest prevalence class in the dataset has a prevalence greater than $\frac{1}{1000}$, then assumption-4 boils down to saying $L < N - 0.5 * \max_y (-\log p(Y = y)) = N - 3.45$, where $L$ is the low PD value caused by spurious features in $D_s$, and $N$ is the total number of layers in the CNN. All our datasets satisfy the class prevalence $> \frac{1}{1000}$ constraint. Even diseases like pneumothorax which are rare, have a class prevalence of at least $\frac{1}{30}$ in both NIH and MIMIC-CXR. And $L < N - 3.45$ is easily satisfied in all our experiments. For e.g., see Fig-5 where 80% (or $1 - \psi = 0.8$) of the samples have PD $\leq 16$. So $L = 16, N = 121$(for Densenet-121) easily satisfies $16 < 121 - 3.45$.

Now we elaborate on the proof of the proposition given the four assumptions. We proceed in two parts: first, we lower bound $\mathcal{V}_{cnn}$-information for $D_s$, and then we upper bound $\mathcal{V}_{cnn}$ for $D_i$.

Assumption 3 implies:

$$(B - A)\tau \leq \sum_{k=A}^{B} \tau \leq \mathcal{I}_{\mathcal{V}_{knn}}^{D}(\phi_k) - \mathcal{I}_{\mathcal{V}_{knn}}^{D}(\phi_{k-1}) \leq (B - A)\epsilon$$

Note: $(A, B)$ are just placeholders for the min and max indices over which the summation is defined. They are replaced by $(L + 1, K)$ and $(K + 1, N)$ below while trying to lower bound $\mathcal{I}_{\mathcal{V}_{cnn}}^{D_s}$ and upper bound $\mathcal{I}_{\mathcal{V}_{cnn}}^{D_i}$ respectively.

**PD - PVI connection.** Note that by definition, when the prediction depth is $k$ for a sample $X$, then $PVI_{knn}(\phi_k(X)) \geq \delta$ but $PVI_{knn}(\phi_{k-1}(X)) < \delta$. This follows from how we compute PD (see Section-3 in the main paper, and Appendix-A.6).

**Lower bounding $\mathcal{I}_{\mathcal{V}_{cnn}}^{D_s}$**

$$
\begin{aligned}
\mathcal{I}_{\mathcal{V}_{cnn}}^{D_s} &= \mathcal{I}_{\mathcal{V}_{cnn \text{ of depth } N-K}}^{D_s}(\phi_K) && \{\text{Assumption-1}\} \\
&\geq \mathcal{I}_{\mathcal{V}_{knn}}^{D_s}(\phi_K) && \{\text{Assumption-2}\} \\
&= \mathcal{I}_{\mathcal{V}_{knn}}^{D_s}(\phi_L) + \sum_{k=L+1}^{K} \mathcal{I}_{\mathcal{V}_{knn}}^{D}(\phi_k) - \mathcal{I}_{\mathcal{V}_{knn}}^{D}(\phi_{k-1}) && \{\text{Telescoping Sum}\} \\
&\geq \mathcal{I}_{\mathcal{V}_{knn}}^{D_s}(\phi_L) + (K - L)\tau && \{\text{Assumption-3}\} \\
&\geq \psi \min_{X,Y \in D_s, pd >= L} PVI_{knn}(X \to Y) \\
&\quad + (1 - \psi) \min_{X,Y \in D_s, pd < L} PVI_{knn}(X \to Y) + (K - L)\tau && \{\text{Prediction Depth Separation}\} \\
&\geq 0 * \psi + \delta * (1 - \psi) + (K - L)\tau && \{\text{Prediction Depth Separation}\}
\end{aligned}
$$

**Upper bounding $\mathcal{I}_{\mathcal{V}_{cnn}}^{D_i}$**

$$
\begin{aligned}
\mathcal{I}_{\mathcal{V}_{cnn}}^{D_i} &\leq \mathcal{I}_{\mathcal{V}_{knn}}^{D_i}(\phi_N) && \{\text{Assumption-2}\} \\
&= \mathcal{I}_{\mathcal{V}_{knn}}^{D}(\phi_K) + \sum_{k=K+1}^{N} \mathcal{I}_{\mathcal{V}_{knn}}^{D}(\phi_k) - \mathcal{I}_{\mathcal{V}_{knn}}^{D}(\phi_{k-1}) && \{\text{Telescoping Sum}\} \\
&\leq \mathcal{I}_{\mathcal{V}_{knn}}^{D}(\phi_K) + (N - K)\epsilon && \{\text{Assumption-3}\} \\
&\leq (N - K)\epsilon + \psi \max_{X,Y \in D_i, pd(X) \leq K} PVI_{\mathcal{V}_{knn}}^{D}(\phi_K(X) \to Y) \\
&\quad + (1 - \psi) \max_{X,Y \in D_i, pd(X) > K} PVI_{\mathcal{V}_{knn}}^{D}(\phi_K(X) \to Y) && \{\text{Prediction Depth Separation}\} \\
&\leq (N - K)\epsilon + \psi \max_{y} (-\log p(Y = y)) \\
&\quad + (1 - \psi) \max_{X,Y \in D_i, pd(X) > K} PVI_{\mathcal{V}_{knn}}^{D}(\phi_K(X) \to Y) && \{\text{PVI} \leq -\log p(Y = y)\} \\
&\leq (N - K)\epsilon + \psi \max_{y} (-\log p(Y = y)) + (1 - \psi)\delta && \{\text{PD-PVI connection for pd} > K\}
\end{aligned}
$$

The proof follows by comparing the lower bound on $\mathcal{I}_{\mathcal{V}_{cnn}}^{D_s}$ and the upper bound on $\mathcal{I}_{\mathcal{V}_{cnn}}^{D_i}$. Intuitively what this means is that when there is a sufficiently large gap in the mean PD between $D_s$ and $D_i$, then the $\mathcal{V}$-information of $D_s$ exceeds the $\mathcal{V}$-information of $D_i$, which is why the model prefers learning the spurious feature instead of using the core features for the task.

$\square$

## A.2   Grad-CAM Visualization

PD plots help us understand the model layers actively used for classifying different images. To further aid our intuition, we visualize the Grad-CAM outputs for the model's arbitrary layer $k$ by attaching a soft-KNN head. Let $g_{knn}$ denote the soft and differentiable version of k-NN. We compute $g_{knn}$ as follows:

$$g_{knn}(\phi_q^k; \phi_{i \in \{1,2,...m\}}^k) = \frac{\sum_{j \in \mathcal{N}(\phi_q^k, 1)} \exp^{-\|\phi_q^k - \phi_j^k\|/s}}{\sum_{j \in \mathcal{N}(\phi_q^k, :)} \exp^{-\|\phi_q^k - \phi_j^k\|/s}}$$

This function makes the KNN differentiable and can be used to compute Grad-CAM (Selvaraju et al., 2017). We use the $\mathcal{L}_1$ norm for all distance computations. $\phi_q^k$ corresponds to feature at layer-$k$ for query image $x_q$. Let $\phi_{i \in \{1,2,...m\}}^k$ be the training data for KNN. Let $\mathcal{N}$ denote the neighborhood function. $\mathcal{N}(\phi_q^k, :)$ returns the indices of K-nearest neighbors for $\phi_q^k$. $\mathcal{N}(\phi_q^k, 1)$ returns indices of images with positive label $(y = 1)$ from the set of K-nearest neighbors for $\phi_q^k$. $s$ is the median for the set of $\mathcal{L}_1$ norms $\{\|\phi_q^k - \phi_j^k\|\}$ for $j \in \mathcal{N}(\phi_q^k, :)$.

## A.3   Vision Experiments

We use the *NICO++* (Non-I.I.D. Image dataset with Contexts) dataset Zhang et al. (2022) to create multiple spurious datasets (Cow vs. Bird; Dog vs. Lizard) such that the context/background is spuriously correlated with the target. NICO++ is a Non-I.I.D image dataset that uses context to differentiate between the test and train distributions. This forms an ideal setup to investigate what spurious correlations the model learns during training. We follow the procedure outlined by Puli et al. (2022) to create datasets with spurious correlations (90% prevalence) in the training data. The test data has the relationship between spurious attributes and the true labels flipped. This is similar to the Chex-MIMIC dataset illustrated in section-4.2. We test our hypothesis using ResNet-18 and VGG16. We train our models for 30 epochs using an Adam optimizer and a base learning rate of 0.01. We choose the best checkpoint using early stopping.

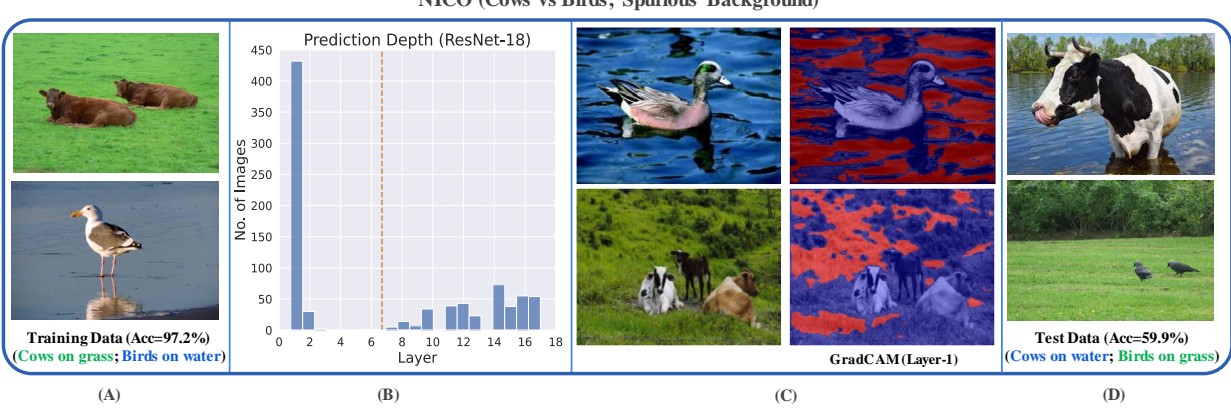

Figure 11: Cow vs. Birds classification on NICO++ dataset. (A) Training data contains images of cows on grass and birds on water (correlation strength=0.9). The model achieves 97.2% training accuracy. (B) PD plot for ResNet-18 reveals a spurious peak at layer-1, indicating the model's heavy reliance on very simple (potentially spurious) features. (C) GradCAM plots for layer 1 reveal that the model mainly relies on the spurious background to make its predictions. (D) Consequently, the model achieves a test accuracy of only 59.9% on test data where the spurious correlation is flipped (i.e., cows (birds) are found on water (grass)).

Figures-11,13 show PD plots and train/test accuracies for models that learn the spurious background feature present in the NICO++ dataset. While all models achieve > 85% training accuracy, they have poor accuracies ( 50%) on the test data where the spurious correlation is flipped. This can be seen simply by observing the PD plots for the model on the training data. The plots are skewed towards the initial layers indicating that the model relies heavily on very simple (potentially spurious) features for the task. GradCAM maps also confirm that the model often focuses on the background context rather than the foreground object of interest.

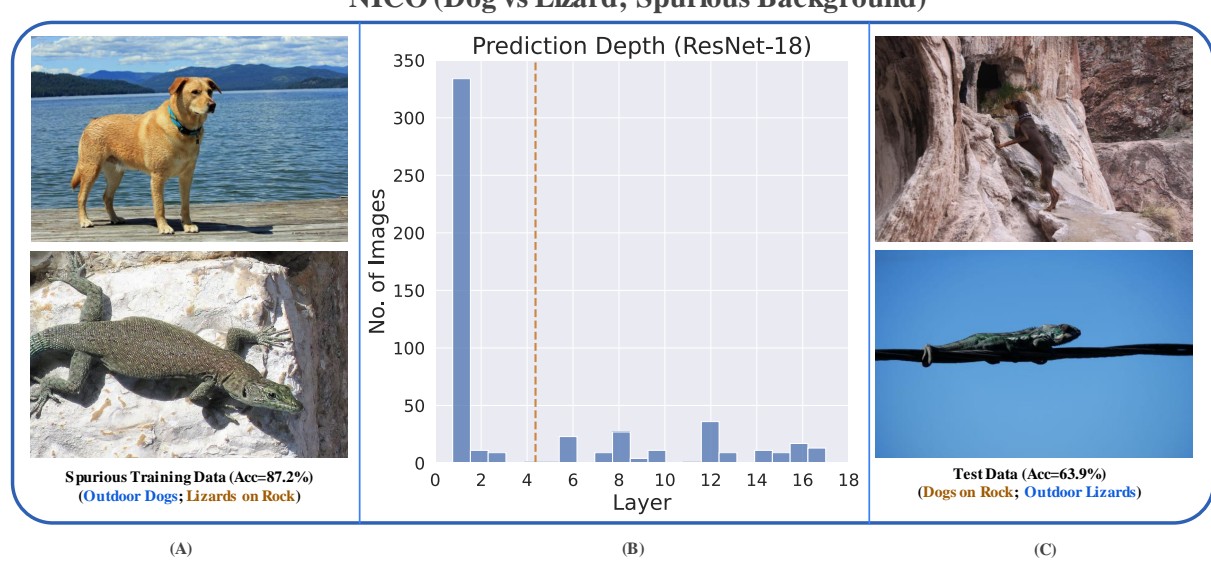

Figure 12: Balanced dataset for Cow vs. Birds classification task on NICO++ dataset. (A) The training dataset contains a balanced distribution of cows and birds found on water and grass (each group has an equal number of images). (B) The balanced dataset shifts the PD plot towards the later layers (compared to Fig-11B, indicating that the model relies lesser on spurious features. (C) This consequently results in an improved test accuracy of 80.2% (as compared to 59.9% in Fig-11D for the spurious dataset).

Figure 13: Dog vs. Lizard classification with a spurious background feature on NICO++ dataset. (A) Training data contains images of outdoor dogs and lizards on rock (correlation strength=0.9). The spurious background color/texture reveals the foreground object. The model achieves 87.2% training accuracy. (B) PD plot for ResNet-18 reveals a spurious peak at layer-1, indicating the model's reliance on simple (potentially spurious) features. (C) The low test accuracy confirms this (63.9%). The test data has the spurious correlation flipped (i.e., images contain dogs on rock and lizards found outdoors.)

We further observe in Fig-12 that balancing the training data (to remove the spurious correlation) results in a model with improved test accuracy (80.2%) as expected. This is also reflected in the PD plot (Fig-12B), where we see that the distribution of the peaks, as well as the mean PD, shift proportionately towards the later layers, indicating that the model now relies lesser on the spurious features.

By monitoring PD plots during training and using suitable visualization techniques, we show that one can obtain useful insights about the spurious correlations that the model may be learning. This can also help the user make an educated guess about the generalization behavior of the model during deployment.

### A.4 Empirical Relationship: PD vs $\mathcal{V}$-information

In Section-4.4 we explore the relationship between PD and $\mathcal{V}$-information. To empirically confirm these results, we further investigate this relationship on four additional datasets: KMNIST, FMNIST, SVHN, and CIFAR10. We train a VGG16 model on these datasets for ten epochs using an Adam optimizer and a base learning rate of 0.01. We use a bar plot to show the correlation between PD and $\mathcal{V}$-entropy. We group PD into intervals of size four and compute the mean $\mathcal{V}$-entropy for samples lying in this PD interval.

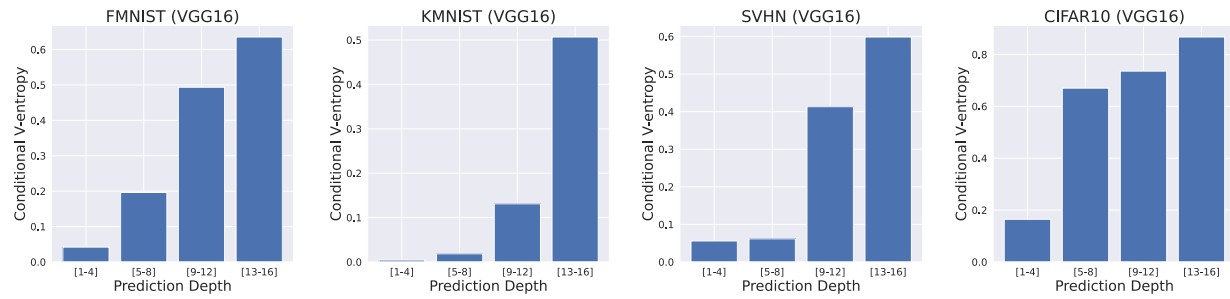

Figure 14: The bar plots show a positive correlation between PD and Conditional $\mathcal{V}$-entropy. Samples with higher PD also have a higher $\mathcal{V}$-entropy resulting in lower usable information for models like VGG16.

In Section-4.4, we find that PD is positively correlated with $\mathcal{V}$-information, and the results shown in Fig-14 further confirm this observation. Instance difficulty increases with PD, and the usable information decreases with an increase in $\mathcal{V}$-entropy. It is, therefore, clear from Fig-14 that samples with a higher difficulty (PD value) have lower usable information, which is not only intuitive but also provides empirical support to Proposition-1 in Appendix-A.1.

### A.5 Chest Drain Annotations for NIH Dataset

To reproduce the results by Oakden-Rayner et al. (2020), we need chest drain annotations for the NIH dataset (Wang et al., 2017a), which is not natively provided. To do this, we use the MIMIC-CXR dataset (Johnson et al., 2019), which has rich meta-data information in radiology reports. We collaborate with radiologists to identify terms related to Pneumothorax from the MIMIC-CXR reports. These include pigtail catheters, pleural tubes, chest tubes, thoracostomy tubes, etc. We collect chest drain annotations for MIMIC-CXR by parsing the reports for these terms using the RadGraph NLP pipeline (Jain et al., 2021). Using these annotations, we train a DenseNet121 model to detect chest drains relevant to Pneumothorax. Finally, we run this trained model on the NIH dataset to obtain the needed chest drain annotations. We use these annotations to get the results shown in Fig - 8D, which closely reproduces the results obtained by Oakden-Rayner et al. (2020).

### A.6 Notion of Undefined Prediction Depth

Section-3 shows how we compute PD in our experiments. While fully trained models give valid PD values, our application requires working with arbitrary deep-learning models that are not necessarily fully trained. We, therefore, introduce the notion of undefined PD by treating $k$-NN predictions close to 0.5 (for a binary classification setting) as invalid. We define a $\delta$ such that $|f_{knn}(x) - 0.5| < \delta$ implies an invalid $k$-NN output. We use $\delta = 0.1$ and $k = 29$ in our experiments. If any $k$-NN predictions for the last three layers are invalid, we treat the PD of the input image to be undefined. To work with high-resolution images (like $512 \times 512$), we downsample the spatial resolution of all training embeddings to $8 \times 8$ before using the $k$-NN classifiers on the intermediate layers. We empirically see that our results are insensitive to $k$ in the range $[5, 30]$.

### A.7  A PD Perspective for Feature Learning

Table 1 shows that the core-only accuracy stays high (>98%) for datasets where the spurious feature is harder to learn than the core feature. When the spurious feature is easier than the core, the model learns to leverage them, and hence the core-only accuracy drops to nearly random chance (∼50%). Interestingly, the KMNpatch-MN results have a much higher core-only accuracy (∼69%) and a more significant standard deviation. This is because the choice of features that the model chooses to learn depends on the PD distributions of the core and spurious features. We provide three different perspectives on why KMNpatch-MN runs have better results.

**PD Distribution Perspective:** The KMNpatch-MN domino dataset has a smaller difference in the core-spurious mean PDs ($2.2 - 1.1 = 1$), as compared to other datasets (for e.g., MN-KMN has a difference of $5 - 2.2 = 2.8$ in their mean PDs). The closer the PD distributions of the core and spurious features are, the more the model treats them equivalently. Therefore, in the case of the KMNpatch-MN, we empirically observe that different initializations (random seeds) lead to different choices the model makes in terms of core or spurious features. This is why the standard deviation of KMNpatch-MN is high (20.03) compared to the other experiments.

**Theoretical Perspective (Proposition-1):** This is not surprising and, in fact, corroborates quite well with Proposition-1 in Appendix-A.1. The Prediction Depth Separation Assumption suggests that without a sufficient gap in the mean PDs of the core and spurious features, one cannot concretely assert anything about their ordinal relationship in terms of their usable information. In other words, spurious features will have higher usable information (for a given model) than the core features only if the spurious features have a sufficiently lower mean PD as compared to the core features. On the other hand, as the core and spurious features become comparable in terms of their difficulty, the model begins to treat them equivalently.

**Loss Landscape Perspective:** *(this is a conjecture; we do not have empirical evidence)* The loss landscape is a function of the model and the dataset. The solutions in the landscape that are reachable by the model depend on the optimizer and the training hyperparameters. Given a model and a set of training hyperparameters, we conjecture that the diversity (in terms of the features that the model learns during training) of the solutions in the landscape increase as the distance (difference in mean PD) between the core and spurious features decreases. This diversity manifests as the model's choice of using core vs. spurious features and could potentially result in a higher standard deviation of core-only accuracy across initializations.

### A.8  Code Reproducibility

The code for this project is publicly available at: `https://github.com/batmanlab/TMLR23_Dynamics_of_Spurious_Features`

### A.9  PD Plot Consistency

We perform several random runs for the experiments shown in Fig-6 and compute the probability density function of the PD plots in Fig-6 using kernel density estimation (KDE). Fig-15 shows the resulting PD envelopes (computed using KDE), and also the original histograms of different random runs.

We can see the consistency of PD plots for any given dataset across runs involving different random seeds. The overall ordering of the datasets according to difficulty computed by mean PD remains the same. This shows that PD can be used as a reliable measure to estimate dataset difficulty and to also detect spurious feature.

### A.10  Ensemble Uncertainty as a Measure of Instance Difficulty

The main idea of our paper can be extended to other instance difficulty metrics and visualization techniques and is not limited to PD or grad-CAM (see Sec-3). We provide a proof of concept in this section by using

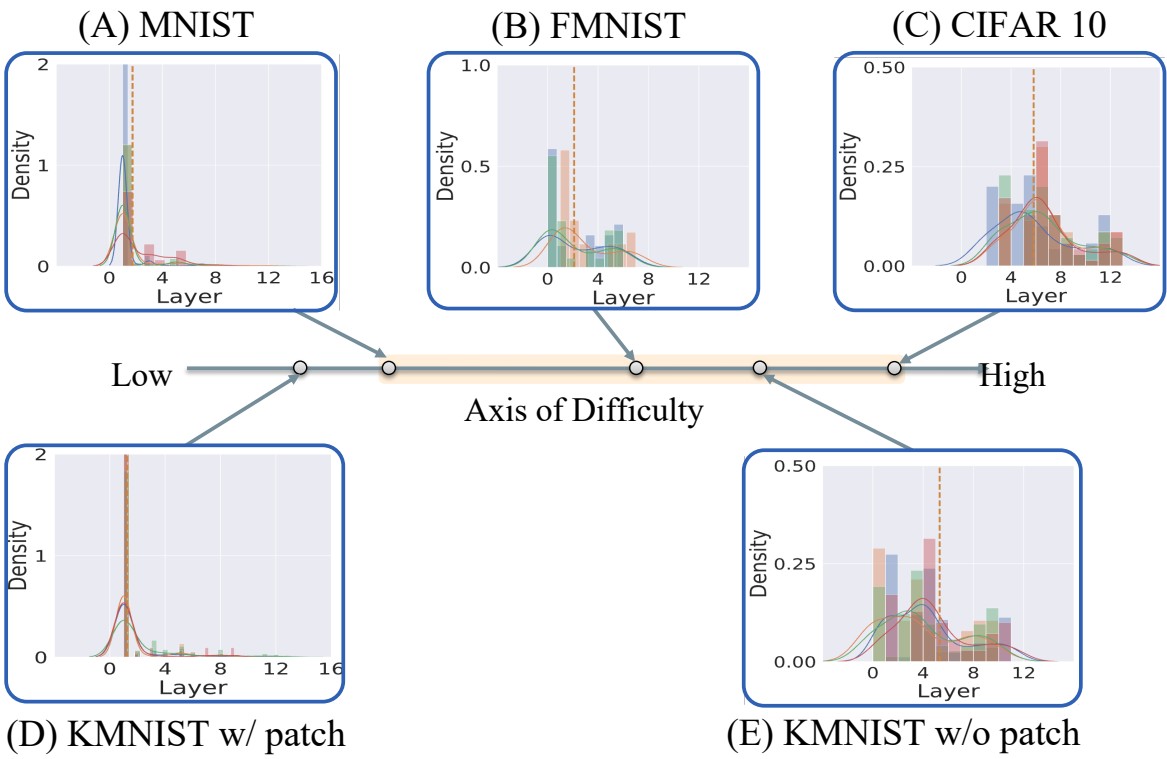

Figure 15: Consistency of PD plots across four random runs. The overall ordering of dataset difficulty based on mean PD remains the same as in Fig-6

techniques different from PD and grad-CAM to detect spurious features. However, there are several added benefits of using sophisticated metrics like PD and we highlight them here.

### A.10.1 Toy Dataset (Synthetic Spurious Feature)

We re-run the experiments shown in Fig-6 with a new set of metrics. In order to compute instance difficulty, we train an ensemble of linear models over the various datasets shown in Fig-6 and compute the uncertainty over softmax outputs for various images. The uncertainty of the ensemble can be used as a proxy for instance difficulty. We compute the uncertainty by taking an expectation over softmax entropy (as common in the literature of uncertainty quantification (Lakshminarayanan et al., 2017; Mukhoti et al., 2023; Van Amersfoort et al., 2020)) of the ensemble models, and we use this metric instead of PD. We order the datasets based on mean entropy, and use SHAP (Lundberg & Lee, 2017) for visualizing the early peaks in order to detect spurious features.

Results are shown in Fig-16. We find that the overall order of the dataset difficulty is the same as in Fig-6, and the entropy plots also look similar to the PD plots computed previously. The spurious patch in the KMNIST dataset significantly decreases the entropy of the ensemble models as expected. We visualize these images using SHAP to find that the model heavily relies on the spurious patch. Upon removing the patch from the KMNIST dataset (see Fig-16E), the mean entropy of the dataset significantly increases as the model now has to rely mainly on the core features. We plot the KMNIST images (with spurious patch) along with their SHAP visualizations for various classes in Fig-17 to show how the model significantly relies on the spurious patch for all the classes.

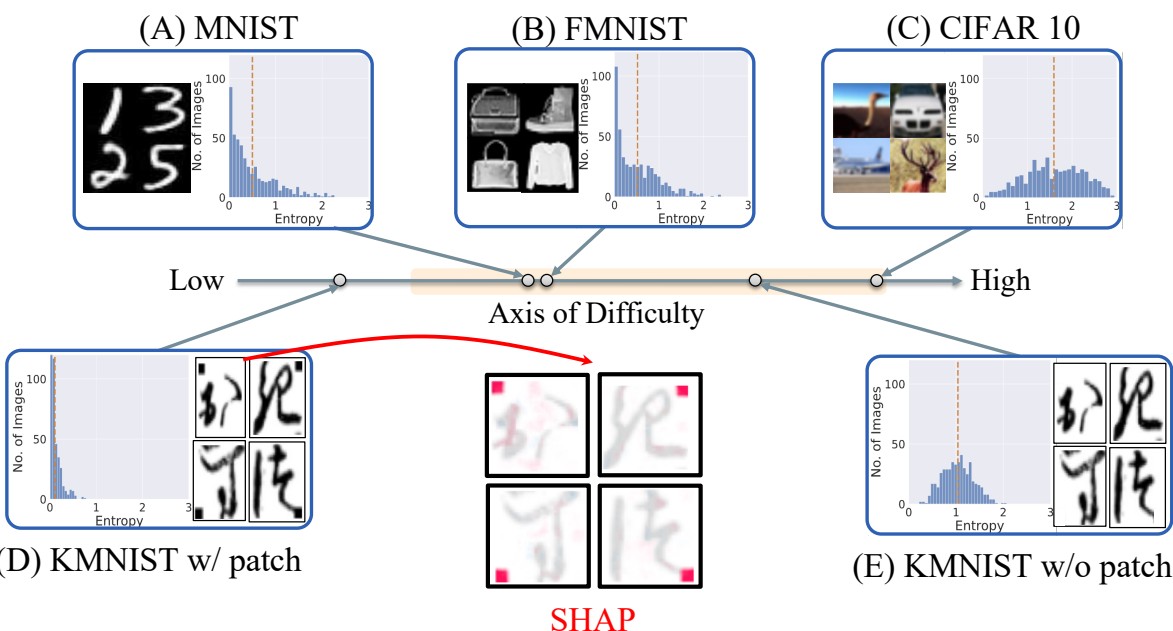

Figure 16: Ensemble uncertainty as a measure of dataset difficulty and SHAP for visualizing spurious features. Top row shows three reference datasets and their corresponding entropy plots. The datasets are ordered based on their difficulty (measured using mean entropy shown by dotted vertical lines): KMN w/ patch(0.107) < MNIST(0.491) < FMNIST(0.527) < KMN w/o patch(1.04) < CIFAR10(1.593). The bottom row shows the effect of the spurious patch on the KMNIST dataset. The yellow region on the axis indicates the expected difficulty of classifying KMNIST. While the original KMNIST lies in the yellow region, the spurious patch significantly reduces the task difficulty. The SHAP plots show that the model focuses on the spurious patch.

### A.10.2 Medical Dataset (Real Spurious Feature)

We now compare the ensemble entropy method with PD on the real medical dataset (NIH) with real spurious features (like chest drain). The setup is exactly the same as in Sec-4.2 (Real Spurious Feature in Medical Dataset). Additionally, we try to detect the simple artifacts and chest drains (as shown in Fig-8) using the ensemble entropy method. We increase the model capacity of the ensemble by adding a convolutional layer and a ReLU layer before the linear classification (Conv–ReLU–Linear). This not only helps the ensemble models to detect more complex spurious features (as compared to simple 1-layer linear models as in the previous section-A.10.1) but also leads to smoother and better grad-CAM plots, which helps us better debug the spurious features the model is using. The setup for PD experiments, however, remains the same as in Sec-4.2.

Fig-18 shows that both the PD and the ensemble entropy method can detect the simple spurious features in the NIH dataset. However, Fig-19 shows that the PD method can additionally detect more complex spurious features like chest drains in the NIH dataset, whereas the ensemble entropy method is not able to do so as it comprises simple convolutional neural networks that have low model-capacity and can therefore only detect simple spurious artifacts.

To further validate if the simple convolutional model used above can learn chest drains, we set up a simple classification experiment to try to classify if the X-ray image in the MIMIC-CXR dataset (Johnson et al., 2019) has chest drains/tubes or not using this simple baseline model. We collect chest drain annotations for the MIMIC-CXR dataset (Johnson et al., 2019) by parsing through radiology reports using the RadGraph NLP pipeline (Jain et al., 2021). We collaborate with radiologists to figure out terms related to Pneumothorax (like pigtail catheters, pleural tubes, chest tubes, thoracostomy tubes, etc.) Using these annotations, we train

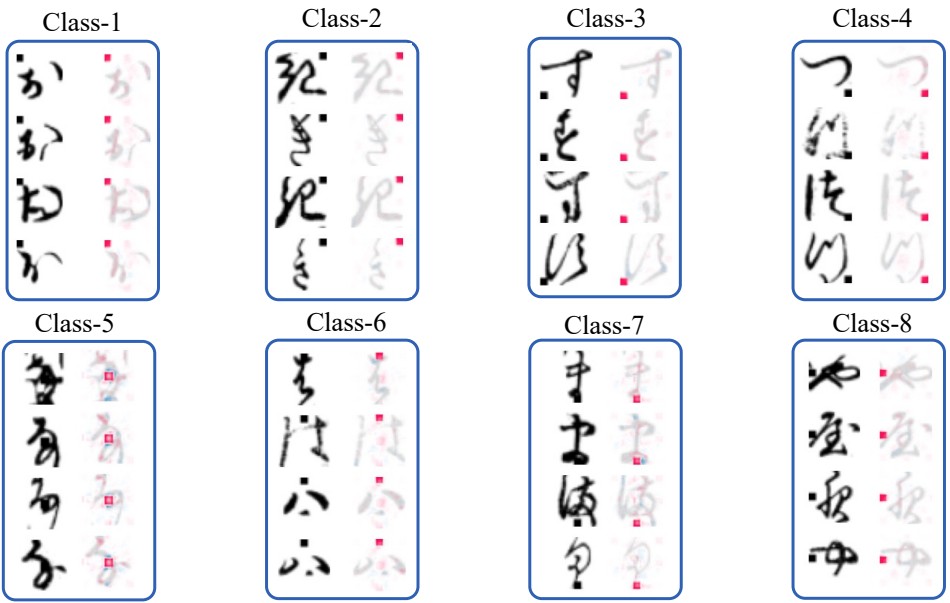

Figure 17: SHAP plots for visualizing what linear models learn on the KMNIST dataset with a spurious patch. The figure shows how the model is significantly relies on the spurious patch for each of the classes shown.

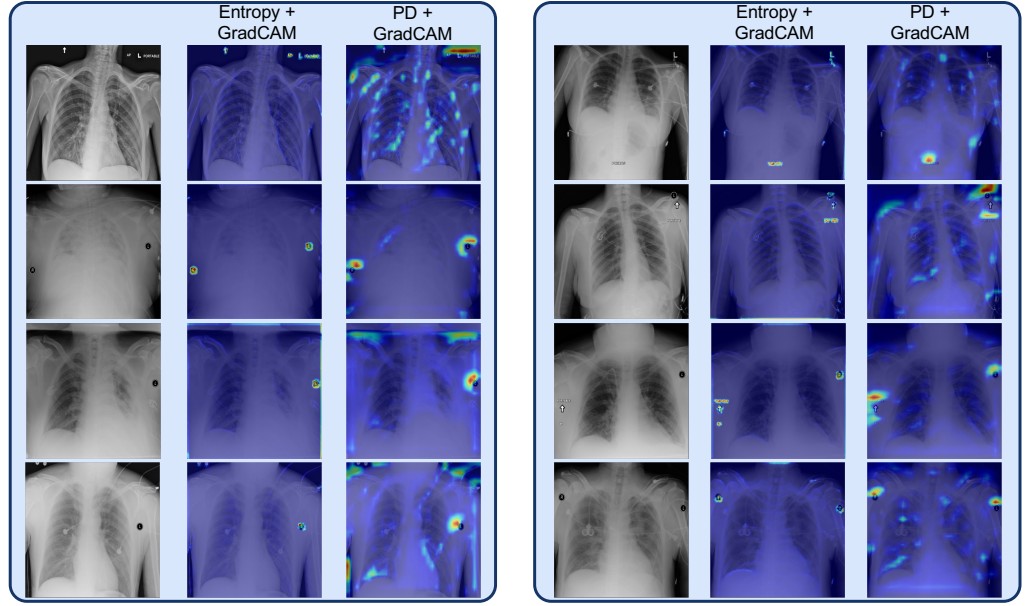

Figure 18: Simple spurious artifacts in NIH dataset. GradCAM plots show that both methods, entropy and PD, can detect simple spurious artifacts in the NIH dataset.

the simple convolutional model for 40 epochs, and the model only achieves an AUC of 0.58 (random guessing gives an AUC of 0.5). This experiment demonstrates that a simple convolutional model is not capable

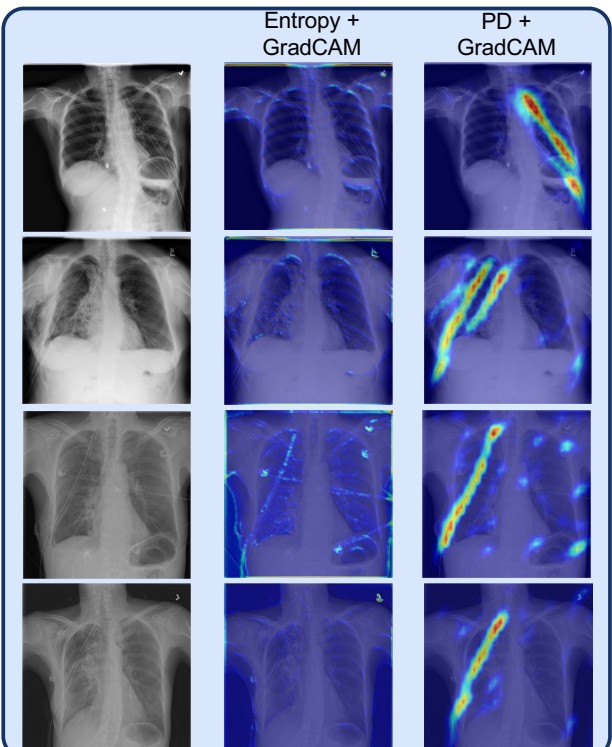 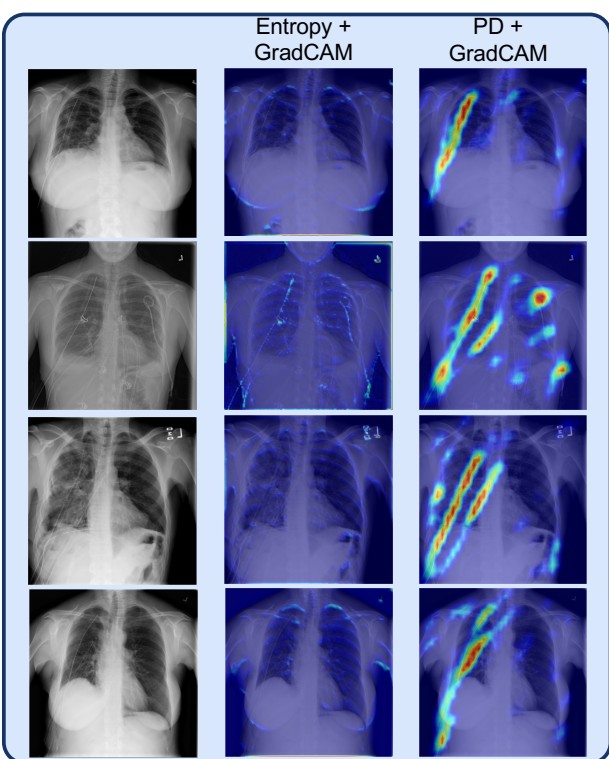

Figure 19: Challenging spurious features (like chest drains) in NIH dataset. Grad-CAM plots for the PD method reveal the spurious chest drain in each of the NIH chest X-ray images. Whereas the entropy method, which comprises an ensemble of simple baseline convolutional architectures, cannot detect more complex spurious features like chest drains. This experiment justifies the use of PD metric over ensemble uncertainty of simple baseline models having low model capacity.

of detecting complex spurious features like chest drains. Therefore, the gradCAM plots for the ensemble entropy method fail to detect chest drains in the NIH dataset (as shown in Fig-19).

This section shows that our work is not limited to PD or GradCAM. It is defined for any example difficulty metric (see Sec-3). We use PD as a proof of concept, but our hypothesis simply suggests that by monitoring the example difficulty of training data points and by visualizing the examples (particularly those with low difficulty), we can detect spurious features that hurt model generalization. The better the example difficulty metric and visualization technique, the better we can detect such spurious features. However, there are several additional benefits of using more sophisticated methods like PD, which are clearly illustrated in this section (see Fig-19). PD measures sample difficulty in a model-specific manner. It takes the model architecture into account, and we show why this is important in Fig-4. The ensemble of linear models approach shown here is not model-specific. It comprises simple linear baseline models and hence cannot detect more challenging spurious features like chest drains. Using sophisticated techniques like PD, one can also attempt to broadly identify the layers responsible for learning spurious features. This can further help develop suitable intervention schemes that remove the spurious feature representations from those layers. This is a promising future direction to address the issue of unlearning spurious features. One may use an ensemble of multi-layer neural networks to compute uncertainty over samples, but training an ensemble of larger models involves significant computational overhead and is time-consuming. Additionally, we develop useful theoretical connections between PD and information-theoretic concepts like usable information (see Appendix-A.1) to explain the empirical success we obtain in our experiments. The above benefits justify our use of the PD metric in this paper.

