# OpenReview forum: "Beyond Distribution Shift: Spurious Features Through the Lens of Training Dynamics"
_TMLR — Accepted by TMLR_

### Review · Reviewer_BeHk · 2023-06-08

**Summary Of Contributions:**

In this study, the authors discovered that spurious features that are harder than the core features are shortcuts. They also found that Prediction Depth serves as an effective metric for detecting spurious features. Moreover, their findings revealed that shortcuts are learnt by early layers and in the early learning phrase. Lastly, the authors underscore the importance of monitoring early training training with Prediction Depth.

**Audience:**

No

**Claims And Evidence:**

No

**Requested Changes:**

1. fonts in some figures are too small to see.


**Strengths And Weaknesses:**

Strengths:

1. The paper's topic holds interest and relevance across multiple domains within machine learning, such as adversarial learning and semi-supervised learning.

2. To validate their hypothesis, the authors conduct experiments on extensive datasets.

Weaknesses:

1. Given that they are termed 'shortcuts', it seems intuitive that spurious features would be less challenging than core features. The inherent suggestion in their names already implies this meaning, so the first claim might not provide particularly meaningful insights.

2. The definition-1 is different from "Shortcuts are spurious features that perform well on standard benchmarks … real-world setting". Standard benchmarks should include test set. Should the distribution deviate substantially from the test set, the results would evidently indicate the necessity to investigate these spurious features further. I think the challenge lies in identifying spurious features that exist in both training and test datasets.

3. The aforementioned issue brings up further questions that challenge the second and third claims made by the authors. Are all spurious features relatively simple patterns, like 'block' used in the experiments? Could there exist spurious features with more complex patterns that remain easier to learn than the core features? If so, these features should be learned in the later layers and during later phases of learning. Therefore, while Prediction Depth (PD) might be useful for identifying evident spurious features, its utility for detecting such features in real-world problems may be limited. For instance, is PD capable of identifying any spurious features in databases such as ImageNet or COCO?

---

> ### Author Response · Authors · 2023-08-18
> **Clarification on definitions, methodology, etc.**
>
> > **1. Given that they are termed 'shortcuts' [...] first claim might not provide particularly meaningful insights.**
>
> We believe the word "shortcut" has led to a lot of confusion in the paper. Kindly see our revised manuscript and all the highlighted changes. What we are trying to say is that spurious features are of two types: *"benign"* (those that do not hurt model generalization) and *"harmful"* (those that hurt model generalization). To detect the harmful spurious features, we use instance difficulty metrics like PD to show that spurious features that are harder than core features are benign, whereas spurious features that are easier than core features are harmful and hurt generalization.
>
> Given a spurious feature, it may not always be apparent whether it is benign or harmful, as that depends on the model architecture, the task, the dataset, etc. The same spurious feature can act benign for one model but harm another. See Fig-4 for an example.  Fig-1 also illustrates how a simple distribution-shift viewpoint framework does not suffice to distinguish between benign and harmful spurious features. The prime number in scenario-1 can be concerning, especially since it perfectly correlates with the label. However, one need not be concerned if the model cannot learn it. So a systematic and mathematically-motivated approach that encompasses and takes into account all these different cases is very much needed to tackle spurious correlations. Our paper is an attempt in this direction.
>
> > **2. [...] Should the distribution deviate substantially from the test set, the results would evidently indicate the necessity to investigate these spurious features further. I think the challenge lies in identifying spurious features that exist in both training and test datasets.**
>
> Thank you for this point!
>
> (a) We agree that if the distribution deviates substantially from the test set, the results will likely indicate the **need** to investigate spurious features further. Our paper **exactly addresses** this need by providing a method to investigate what spurious features are being learned by the model and at what layer they are being learned.
>
> (b) Additionally, our method allows for the **early detection** of these spurious features in most cases (see Sec-4.3, Fig-9&10). So one need not wait until the model is fully trained for further investigation.
>
> (c) A subtle caveat here is that a significant distribution shift between train and test may not always be reflected in the results because the distribution shift viewpoint cannot distinguish between benign and harmful spurious features (see Fig-1). We incorporate this scenario in our definitions by including various factors like feature difficulty, model, etc., while classifying spurious features (see Sec-3).
>
> (d) Our method helps us detect features learned by the various layers of the model and hence is applicable even in scenarios where a feature exists in both training and test datasets. Note that a feature that is correlated with the label in both train and test data cannot be termed *"spurious"* as per the general definition of *"spurious"* in the ML literature. Please refer to Sec-2 (Related Work), which includes definitions given by Geirhos et al. (2020), Wiles et al. (2021), Bellamy et al. (2022), etc. Since test accuracy is what matters, features correlated with labels in train and test data do not hurt generalization, so one need not worry about them.
>
> > **3. [...] Are all spurious features relatively simple patterns, [...] while Prediction Depth (PD) might be useful for identifying evident spurious features, its utility for detecting such features in real-world problems may be limited [...]**
>
> (a) We humbly disagree. We are not claiming that **all** spurious features are just simple patterns. Many of them can be more complex and hence may be detected in the later layers of the model. PD can actually do well even in such situations, and that is why we include real-world examples like NIH, which have real spurious features like chestdrains which are not as simple as detecting a white patch or block. In fact, in Fig-8 we show two kinds of spurious features. The simple artifacts shown in Fig-8b require only 1-4 layers for the model to learn, but some of the chestdrains shown in Fig-8c were detected in layer-40. So PD is capable of detecting more complex patterns.
>
> (b) Secondly, PD is not a perfect metric and has limitations. However, our workflow is **generic** and **not limited to PD or GradCAM**. It is defined for any example difficulty metric (see Sec-3). We use PD as a proof of concept, but our hypothesis simply suggests that by monitoring the example difficulty of training data points and by visualizing the examples (particularly those with low difficulty), we can detect spurious features that hurt model generalization. The better the example difficulty metric and visualization technique, the better we will be able to detect such spurious features.

---

> > ### Comment · Reviewer_BeHk · 2023-09-08
> > **Thank you for your reponse!**
> >
> > Thanks for the authors' response. Most of my concerns have been addressed.

---

### Review · Reviewer_vHSj · 2023-06-08

**Summary Of Contributions:**

The paper develops a distinction between shortcuts and spurious correlations in general, by emphasizing the dependence of shortcuts on learnability. The authors present empirical and theoretical analyses to support these claims. To formalize learnability, the authors start by introducing the notion of Prediction Depth (PD), which states an example is easier if it can be classified by the earlier embedding layers in a neural network. From here, the authors state that a dataset or task is easier if its mean PD is lower. Obviously, these definitions depend intricately on the model class.

The connection back to features from PD is given by examining the histogram of PDs over the dataset. The authors argue that a peak in this histogram at earlier layers suggests the existence of shortcut features. By examining these histograms over the course of training, the authors claim these shortcut features are learned early in training since the peaks in the histogram at early layers remains constant. Finally a theoretical connection between PD and V-usable information is made.

**Audience:**

Yes

**Broader Impact Concerns:**

None.

**Claims And Evidence:**

No

**Requested Changes:**

I feel the paper needs significant rewriting. As discussed above in the Strengths and Weaknesses section, there are major difficulties with the definitions used. These lead to potentially circular claims and an analysis of results that I found difficult to follow.

**Strengths And Weaknesses:**

**Strengths**
- Overall the experiments are quite interesting and have a reasonable mix of datasets.
- The figures present the papers result reasonably well and are easier to follow than the text.
- The general point that it is important to consider the learnability of spurious features is valid.

**Weaknesses**
- **Lack of clarity in definitions and analysis.** The authors make the following definition of shortcuts: spurious features that are easy to learn. Unfortunately this definition rests crucial on the definition of easiness to learn and is highly model dependent.
- **PD as definition of difficulty.** First, this definition has no dependence on the labels, which seems odd. Even trivial models could return a PD of 1. For example, imagine that the first embedding layer is just a constant function. This would be true regardless of how intuitively “difficult” a task is. The inline equation is also confusing. There is a minimization and maximization, but only one variable (n) is used. This should be rewritten, since you want the minimum n that maximizes the function.
- **Model dependence.** PD depends intricately on the architecture of the neural network. Indeed, it seems highly likely that changing the architecture could lead to contradictory results, where two dataset have a different ordering on their mean PD for two different architectures.
- **Poorly written.** The analysis is quite hard to follow and contains many circular claims. For example, the line “This experiment demonstrates that not all spurious correlations are shortcuts, but only those spurious features that are easier than the core features are potential shortcuts.” Under the authors own terms, this is just the definition of a shortcut and not something that should be empirically tested.
- **Definition 2.** This is not a definition, it simply introduces notation. In particular, “difficulty of predicting” is not defined and neither is “model.” What does “model” refer to here? Is this a model class, a trained model, etc?

---

> ### Author Response · Authors · 2023-08-18
> **Clarifying the definitions and analysis [Part-1/2]**
>
> We are sorry for the confusion caused. We have made **significant changes** to the manuscript to improve clarity. Kindly look at the revised manuscript and the general comment we posted above titled: "Summary of the major changes to the paper".
>
> > **Lack of clarity in definitions and analysis: The authors make the following definition of shortcuts: spurious features that are easy to learn. Unfortunately this definition rests crucial on the definition of easiness to learn and is highly model dependent.**
>
> The revised manuscript and general comment above should clarify this to great degree. Our definition of harmful spurious feature rests on the definition of easiness to learn, and this is **highly model dependent**. This is **exactly** the point we are trying to make in the paper. Whether a spurious feature is *"benign"* or *"harmful"* very much depends on the model. A spurious feature may be benign for one model but harmful for another. To make this more clear, we have added Figure-4 in the revised manuscript. This figure illustrates that the spurious white patch is benign for Deep Sets model but harmful for MLPs. So the model must be taken into account.
>
> The main point of the paper, as hinted in the title as well, is that the "distribution shift" viewpoint is limited because it only looks at the distribution of train and test data and ignores other important aspects like feature difficulty, model, etc. Therefore the distribution shift viewpoint cannot distinguish between benign and harmful spurious features (see Fig-1). Whereas our definitions (see Sec-3) incorporate these aspects, using which we show that not all spurious features hurt generalization.
>
> > **PD as definition of difficulty: First, this definition has no dependence on the labels, which seems odd. [...] The inline equation is also confusing [...] This should be rewritten, [...]**
>
> That's a good observation! Actually, it is not very odd that the PD does not depend on the label of the given input, and it is an interesting point you bring up. The purpose of PD is to *measure the difficulty* of the input as perceived by the model. It is *not measuring accuracy* or how well the model is able to classify the input. Measures similar to accuracy (like AUC, AUPRC....) all take into account the label. Whereas PD simply says how many layers the model needs to classify the input into a particular class (even if that's the wrong class). If the model is wrongly (rightly) classifying the input, the PD is simply telling how many layers the model needs to wrongly (rightly) classify the input.
>
> Secondly, the $k$-NN classifiers at every layer definitely take into account the labels of the rest of the training data in order to classify the feature embeddings of input $x$ at that layer. We do not show the training data and their labels in Eq-1 to avoid clutter in notation. Fig-2 visually illustrates all this.
>
> We have updated equation-1 and added Fig-2 in the paper to make the PD definition more clear (the general comment explains this in detail). We hope this clarifies how PD is computed.
>
> > **Model dependence: PD depends intricately on the architecture of the neural network. Indeed, it seems highly likely that changing the architecture could lead to contradictory results, where two dataset have a different ordering on their mean PD for two different architectures.**
>
> Thank you for this good point! **PD definitely depends on the model architecture, and that is exactly why we are using it.** Instance difficulty metrics that do not take the model architecture into account are limited because they cannot explain why different models learn different spurious features, and why some spurious features are benign for some models but harmful for other models. To explain these facts requires usage of metrics like PD which take the model architecture into account. Changing the architecture can definitely change the ordering of the mean PD, but this is not a contradiction. It simply explains why different models can potentially learn different spurious and core features. To explain all this more clearly we add Fig-4 which clearly illustrates how MLPs can easily learn the location of spurious white patch, but Deep Sets cannot do so because of the way they process the input.

---

> ### Author Response · Authors · 2023-08-18
> **Clarifying the definitions and analysis [Part-2/2]**
>
> > **Poorly written: The analysis is quite hard to follow and contains many circular claims. For example, the line “This experiment demonstrates that not all spurious correlations are shortcuts, but only those spurious features that are easier than the core features are potential shortcuts.” Under the authors own terms, this is just the definition of a shortcut and not something that should be empirically tested.**
>
> Sorry for the confusion. Please look at our general comment where we explain in detail how our work and definitions do not lead to circular claims. The term "shortcut" is particularly creating the confusion. Our hypothesis is that "Not all spurious features hurt generalization!" (or) "Not all spurious features are harmful!". We use instance difficulty metrics like PD to empirically show that only when the spurious feature is easier than the core, it hurts the model generalization. When the spurious features have higher PD than core features, the model ignores them and hence the generalization is not affected.
>
> > **Definition 2: This is not a definition, it simply introduces notation. In particular, “difficulty of predicting” is not defined and neither is “model.” What does “model” refer to here? Is this a model class, a trained model, etc?**
>
> To clarify the notion of task difficulty, we have re-organized the manuscript to include two immediate examples for measuring task difficulty $\Psi_{\mathcal{M}}^{P}$. How we define/measure or quantify "difficulty of predicting label $y$ from input $X$" depends on the choice of metric. For e.g., PD quantifies this by computing the number of layers needed by the model to classify the input. The model $\mathcal{M}$ used in the notation refers to a function that maps the input $X$ to label $y$. For PD, $\mathcal{M}$  refers particularly to convolutional architectures. $\mathcal{M}$  can be trained or untrained. For $\mathcal{V}$-Usable Information, $\mathcal{M}$ must be a trained model.
>
> We hope these examples and the revised manuscript clarify all your questions to great degree.

---

### Review · Reviewer_4fZW · 2023-07-31

**Summary Of Contributions:**

This work investigates *shortcuts*, which describe a particular kind of spurious
correlation which dominates the prediction for models over class-relevant features.
Based on the statement of related work that "all spurious correlations are
shortcuts", the work shows that spurious correlations do not qualify as
*shortcuts* when they do not provide a pattern which is easier to learn than
the class-relevant features.
A method from previous work (Prediction Depth) is used as a proof-of-concept to
empirically quantify the instance difficulty over datasets.
Using PD, the work empirically shows that "easy" features are learned within
the first two epochs of the training.
A formal connection between PD and v-usable information from another previous
work is demonstrated, backed with some empirical results.
This work highlights the importance to monitor the instance difficulty over datasets to detect shortcuts.

**Audience:**

Yes

**Broader Impact Concerns:**

None.

**Claims And Evidence:**

No

**Requested Changes:**

- (critical) The work must report the number of trials and error bars to
  support the claim of statistical significance.

- (critical) A linear baseline (similar to He et al. 2019) to quantify the
  instance difficulty should be used to provide an additional, even simpler
  instance difficulty metric (and justify the use of the more complex
  PD-metric).

- (non-critical) The class-relevant locality required for the assumption of kNN
  in higher layer feature space would be helpful if discussed.

- (non-critical) A discussion on the case where class-relevant features of
  various difficulties may be an interesting addition.

### Misc. (non-critical):

- While I am happy the paper comes with code, for the sake of reproducibility I
  would urge the authors to structure their code as a Python module using
  `setuptools`, without relying too heavily on jupyter notebooks. Adhering to
  some code styling (PEP8, e.g. via black) and keeping to some docstring-style
  (googledoc, numpydoc, etc.) does wonders for readability and will thus boost
  reproducibility and accelerate subsequent research.

- (figure-4) (sections-4.2,A.3) looks somewhat odd

- Def2: Let ... indicate~~s~~

- The text captions of the PD figures is a little too small in all figures after Fig. 2


**Strengths And Weaknesses:**

### Strengths

- The paper is very well written, polished and an enjoyable read.

- The figures are of high quality, and highlight the results well.

- The difficulty of various datasets, mostly synthetic, with one real-world
  experiment, is demonstrated very well.

- Given the multiple recent works claiming all spurious correlations are
  supposedly *shortcuts*, this work challenges those over-hasty assumptions,
  which is an important contribution to the field.

- The work motivates the important habit of checking for *shortcuts*, which is
  not reflected in the test accuracy alone.

### Weaknesses

- (critical) The work does not report error bars or number of trials. Reporting
  these is critical to ensure the statistical significance of the empirical
  experiments.

- (critical) The work verifies the hypothesis using the (somewhat complex)
  prediction depth. In the experiments, the main indicator for *shortcuts* is
  the number of samples with a PD of 1, i.e., the number of samples that can be
  classified with kNN after a single linear transformation (plus non-linear
  activation?). The empirical results would be stronger with a
  weaker, linear baseline to measure the instance difficulty, somewhat similar
  to He et. al (2019), which will most likely also find the static artifact
  shortcuts in the synthetic toy datasets.

- (non-critical) The Prediction Depth assumes that the data points are
  transformed in feature space such that they lie close when they will
  ultimately be classified as the same class. For the first (linear) layer and
  simple static artifacts, this may be (intuitively) true. There is no
  guarantee that this is true for higher layers. This is arguably an issue of
  PD, which is used as a proof-of-concept approach to measure instance
  difficulty. Given that this is the only approach used (so far), it would be
  helpful if this would be discussed, maybe in the Appendix A.1.

- (non-critical) Definition 3 defines spurious correlations as shortcuts if
  feature $s$ is strictly *easier* than the class-feature. This definition
  becomes somewhat challenging when assuming that classes may be activated by
  features of various difficulties, where the spurious feature lies within the
  least and most difficult class-defining features. This is somewhat going into
  the direction of the transposed statement "Not all shortcuts are spurious
  correlations" (this is obviously not true given the definition of shortcuts
  in this work).

---

> ### Author Response · Authors · 2023-08-18
> **Thank you for your comments! Seeking a few clarifications.**
>
> We are very glad that you found our work interesting and that it was an enjoyable read for you.
>
> We are running your experiments and making the requested changes (including the code reproducibility suggestions). We have two clarifications.
>
> > **(critical) A linear baseline (similar to He et al. 2019) to quantify the instance difficulty should be used to provide an additional, even simpler instance difficulty metric (and justify the use of the more complex PD-metric).**
>
> Based on the cited paper and your comments, it seems like you want us to replace the $k$-NN classifiers used in the embedding layers of the trained model with a simple linear classifier and then compute PD using these simple baseline models at each layer. Is this accurate?
>
> > **(critical) The work must report the number of trials and error bars to support the claim of statistical significance.**
>
> Table-1 reports the mean and standard deviation for all experiments. All the other results in the paper show PD plot distributions which are histograms of example difficulty for images in the training data. It is not possible to compute error bars over each of the bins of the PD plot because each time we randomize the seed or experiment, it will result in a slightly different PD plot distribution. What we can do is visualize several PD plot distributions in one plot, which shows how the envelope of the various PD plots looks like across runs. Is this what you want us to do?
>
> Please let us know, and we'll update the manuscript. Thank you!

---

> > ### Comment · Reviewer_4fZW · 2023-08-21
> > **Thank you for your rebuttal!**
> >
> > I thank the authors for their clarifications to the other reviewers and their updated manuscript. The updated manuscript is significantly stronger.
> >
> > >> **(critical) A linear baseline (similar to He et al. 2019) to quantify the instance difficulty should be used to provide an additional, even simpler instance difficulty metric (and justify the use of the more complex PD-metric).**
> >
> > My idea writing this response was that since PD measures the sample-difficulty with respect to the model architecture, as clarified in your response to **Reviewer vHSj**, and given that the spurious correlations presented in the manuscript seem to mostly be locate-able in the input, was that one could compare PD with a much simpler approach, with a simpler definition of *sample difficulty*: measure the difficulty using the performance of a much simpler (not necessarily, but possibly linear) model.
> >
> > While this does depend on the labels, and not on the architecture (you motivate that *simple* depends on the model), this is precisely why this would be a good baseline that can be leveraged to justify the use of the more complex PD-metric. Since the accuracy of a single linear classifier could be noisy, one possible way to measure the difficulty would be to train an **ensemble of multinomial logistic regression** (or single-layer MLPs) and compute the **uncertainty of each sample over the ensemble**. Samples with low uncertainties will be *easy* given that a simple model architecture classifies them correctly with low uncertainty.
> >
> > >> **(critical) The work must report the number of trials and error bars to support the claim of statistical significance.**
> >
> > Yes, the PD plots were precisely my concern. Showing several PD plot distributions would solve this issue.

---

> > > ### Author Response · Authors · 2023-08-22
> > > **Thank you for your suggestions!**
> > >
> > > Thank you for your excellent suggestions! We have incorporated these changes **(see Appendix A.9 and A.10)** which significantly improve our manuscript.
> > >
> > > > **(critical) A linear baseline (similar to He et al. 2019) to quantify the instance difficulty should be used to provide an additional, even simpler instance difficulty metric (and justify the use of the more complex PD-metric).**
> > >
> > > We use section A.10 to address this concern and also showcase how our work is generic and not limited to just PD and Grad-CAM. Additionally, we justify why PD is a better metric than the suggested simpler alternatives and hence the heavy usage of PD in our main experiments.
> > >
> > > We compute ensemble uncertainty by taking an expectation over the softmax outputs of various (five) linear baseline models trained on datasets shown in Fig-16. This approach is common in the literature *(refer to Lakshminarayanan et al., 2017; Mukhoti et al., 2023; Van Amersfoort et al., 2020)*. We order the datasets based on mean entropy and use SHAP *(Lundberg & Lee, 2017)* for visualizing the early peaks to detect spurious features. We find that the overall order of the dataset difficulty remains the same, and the entropy plots also look similar to the PD plots computed previously. The spurious patch in the KMNIST dataset significantly decreases the entropy as expected, and SHAP visualization reveals the spurious patch.
> > >
> > > The **main messages of this experiment** are as follows:
> > >
> > > (1) **Our work is not limited to PD and GradCAM**, and one can obtain similar insights by using other metrics (like ensemble uncertainty instead of PD and SHAP instead of GradCAM)
> > >
> > > (2) A **simple approach** involving linear baseline models **can reveal valuable insights**. This can be a quick sanity check approach to check for artifacts in the dataset.
> > >
> > > (3) **Main drawback** here (as you point out) is that this approach is **not model-specific** (Fig-4 visually illustrates why this is important). Whereas, PD takes the model architecture into account. Sec-3 explains that the **task difficulty metric must depend on the model $\mathcal{M}$** as different models can potentially learn different core/spurious features.
> > >
> > > (4) Sophisticated techniques like **PD can broadly identify the layers responsible for learning spurious features**. This can help develop intervention schemes that remove spurious feature representations from those layers (a promising future direction).
> > >
> > > (5) Also, **simple linear models can only fit toy datasets** like MNIST but will not be able to learn and generalize on larger datasets with high-resolution images like Chexpert, MIMIC, NIH, NICO, etc. One may use an ensemble of multi-layer neural networks to compute uncertainty over samples, but this is time-consuming and involves significant computational overhead.
> > >
> > > (6) Additionally, we develop useful **theoretical connections between PD and information-theoretic concepts** like usable information (see
> > > Appendix A.1) which explains the empirical success we obtain in our experiments.
> > >
> > > **The above benefits justify our use of the PD metric in this paper.**
> > >
> > > > **(critical) The work must report the number of trials and error bars to support the claim of statistical significance.**
> > >
> > > **Appendix A.9 shows the consistency of PD plots.** We perform four random runs for the experiments shown in Fig-6 and compute the probability density function of the PD plots using kernel density estimation (KDE). Fig-15 shows the resulting PD envelopes (computed using KDE) and also the original histograms of different random runs. We can see the consistency of PD plots for any given dataset across runs involving different random seeds. The overall ordering of the datasets according to difficulty computed by mean PD remains the same.
> > >
> > > **This shows that PD can be used as a reliable measure to estimate dataset difficulty and detect spurious features.**

---

> > > > ### Comment · Reviewer_4fZW · 2023-08-22
> > > > **A few more points about the PD-metric justification**
> > > >
> > > > Thank you for these swift new results!
> > > >
> > > > I am satisfied with Appendix A.9.
> > > >
> > > > For A.10, I have a few more points I would like to note:
> > > >
> > > > - You mention in the text that *these simple, linear models can only fit toy datasets and will not be able to learn and generalize on larger datasets*, which is exactly the reason why we would like to use these models: they do not generalize, and thus can only rely on simple features, such as the spurious correlations we are trying to detect. I think the argument that they do not generalize is somewhat confusing. I assume what you are trying to say is that these models are not strong enough to find spurious correlations more complex than patches and such (a single convolutional layer might be a good choice)? It should be made clear that, rather than generalizing to the data set, the simple model's purpose is to be only as strong as the spurious feature we are trying to find requires.
> > > >
> > > > -  Given the above, I would expect that we can still find the spurious correlations in Chexpert, MIMIC, NIH, NICO etc.. Of course, the model performance will be much lower, but that does not matter, since we are trying to solve a task that should be impossible with the model, and only be possible through the use of spurious features (i.e., the amount of samples that have a low uncertainty even though the task should be impossible). Did you try these datasets, or only state that the models *do not generalize*?
> > > >
> > > > - The newly added A.10 is not directly compared to PD. The *justification claim* currently only relies on the fact that PD is architecture-aware. I think the claim would be supported much stronger with a direct comparison, like an experiment showing what does not work with the simple approach, but does with PD.

---

> > > > > ### Author Response · Authors · 2023-08-22
> > > > > **Thank you for the clarification!**
> > > > >
> > > > > This greatly clarifies what you have been asking for. I think we now understand it better.
> > > > >
> > > > > Yes, that would be a very interesting experiment indeed and we're curious to see how they compare with PD. This will also nicely justify the use of PD.
> > > > >
> > > > > We're working on these experiments, and we will try to get back to you at the earliest.

---

> > > > > ### Author Response · Authors · 2023-08-25
> > > > > **Additional Experiments to Justify the PD metric**
> > > > >
> > > > > Dear Reviewer 4fZW, we are sorry for the delay. We have run the experiments you have requested and also some additional experiments to justify the observations. Please see the updated A.10 section (we are pasting it below for your convenience).
> > > > >
> > > > > We compare the ensemble entropy method with PD on the real medical dataset (NIH) with real spurious features (like chest drain). The setup is exactly the same as in Sec-4.2 (Real Spurious Feature in Medical Dataset). Additionally, we try to detect the simple artifacts and chest drains (as shown in Fig. 8) using the ensemble entropy method. We increase the model capacity of the ensemble by adding a convolutional layer and a ReLU layer before the linear classification (Conv--ReLU--Linear). This not only helps the ensemble models to detect more complex spurious features (as compared to simple 1-layer linear models as in the previous section-A.10.1) but also leads to smoother and better grad-CAM plots, which helps us better debug the spurious features the model is using. The setup for PD experiments, however, remains the same as in Sec-4.2.
> > > > >
> > > > > **Fig-18 shows that both the PD and the ensemble entropy method can detect the simple spurious features in the NIH dataset. However, Fig-19 shows that the PD method can additionally detect more complex spurious features like chest drains in the NIH dataset, whereas the ensemble entropy method is not able to do so as it comprises simple convolutional neural networks that have low model-capacity and can therefore only detect simple spurious artifacts.**
> > > > >
> > > > > **To further validate if the simple convolutional model used above can learn chest drains, we set up a simple classification experiment to try to classify if the X-ray image in the MIMIC-CXR dataset has chest drains/tubes or not using this simple baseline model.** We collect chest drain annotations for the MIMIC-CXR dataset by parsing through radiology reports using the RadGraph NLP pipeline. We collaborate with radiologists to figure out terms related to Pneumothorax (like pigtail catheters, pleural tubes, chest tubes, thoracostomy tubes, etc.) Using these annotations, **we train the simple convolutional model for 40 epochs, and the model only achieves an AUC of 0.58 (random guessing gives an AUC of 0.50).** This experiment demonstrates that a simple convolutional model is not capable of detecting complex spurious features like chest drains. Therefore, the gradCAM plots for the ensemble entropy method fail to detect chest drains in the NIH dataset (as shown in Fig-19).
> > > > >
> > > > > **Please let us know if this addresses your concerns?** Thank you!

---

> > > > > > ### Comment · Reviewer_4fZW · 2023-09-01
> > > > > > **Satisfied with the new additions**
> > > > > >
> > > > > > Dear authors,
> > > > > >
> > > > > > I apologize for my delayed response.
> > > > > > Thank you for these new results. These analyses nicely show the limitations of the simple method to justify the use of PD in the real world case.
> > > > > > I have no more questions and am satisfied with the updated state of the manuscript, although a reference in the main part to these experiments may be good (which I might have missed).

---

> > > > > > > ### Author Response · Authors · 2023-09-01
> > > > > > > **Thanks to you for your very nice suggestions!**
> > > > > > >
> > > > > > > Dear Reviewer,
> > > > > > >
> > > > > > > Thank you very much! We're very glad it addressed your concerns.
> > > > > > >
> > > > > > > And we thank you for these nice suggestions which helped us improve the paper a lot! Engaging in this discussion actually gave us a much better perspective and understanding of our own work. If not for your suggestions and comments, we wouldn't have come up with such a simple yet insightful experiment to justify the PD metric!
> > > > > > >
> > > > > > > We will surely add more references to this in the manuscript.
> > > > > > >
> > > > > > > Warm Regards,
> > > > > > >
> > > > > > > The Authors

---

> ### Author Response · Authors · 2023-08-22
> **Response to the non-critical suggestions**
>
> > **(non-critical) The class-relevant locality required for the assumption of kNN in higher layer feature space would be helpful if discussed.**
>
> This is a good point. Convolutional neural networks pass the original input through a set of transformations until the features in the last layer can be class-separated by a simple linear layer. This suggests that the class-specific features become more and more linearly separable as we go deeper into the model. A good accuracy or AUC suggests that the features in the last layer are linearly well-separated, but this is not necessarily true for the intermediate layers (as you hint at). However, $k$-NNs (used in the intermediate layers during PD computation) only make assumptions on the *local cohesiveness* of classes (the *local neighborhood* of any point in the feature space comprises highly *"similar features"*) rather than a global assumption on linear separability between features. This local assumption is much looser (less strict) than a global linear separation and given that the classes are eventually even linearly separable in the last layer, a much looser assumption like class-relevant locality may hold true for most of the intermediate layers.
>
> Our evidence for this class-relevant locality assumption in the intermediate layers is more empirical than theoretical. In all our experiments, we could clearly compute dataset difficulty and compare between datasets and other metrics like PVI, and we find good agreement in all these experiments. This empirically proves that the class-relevant locality assumption must hold in most cases; otherwise, our empirical results wouldn't have shown success. The original PD paper (Baldock et al., 2021) shows many more experiments using PD, and their empirical success also indicates that this assumption is true to a good extent.
>
> We will add this discussion in the appendix as you suggest.
>
> > **(non-critical) A discussion on the case where class-relevant features of various difficulties may be an interesting addition.**
>
> This is a very good point and can lead to a whole separate paper in itself. We are not aware of any in-depth study regarding this question (partly because of the challenging nature of the problem). Different images have different core/spurious features, and different models learn different sets of features to different degrees. The optimization algorithm and other training hyper-parameters can also subtly affect what kind of features the model learns. We only have a general/broad understanding that the model simply learns the easiest set of features (core or spurious). This is easy to analyze, particularly when there is a high correlation between a certain set of features (spurious or core) and the label. E.g., Scimeca et al. (2021) assume ideal conditions of perfect correlation between spurious features and labels to show that in the presence of multiple spurious features, the model will learn the "easiest" one. They use approximations to Kolmogorov complexity to quantify "easy". Shah et al. (2020) have previously hinted at this by saying that DNNs are biased toward simple solutions. Later works like Dagaev et al. (2021) come up with the "too-good-to-be-true" prior, highlighting that simple features/solutions do not generalize well. These papers give a general understanding of your question, but a more conclusive answer requires a more in-depth study!
>
> > **The text captions of the PD figures is a little too small in all figures after Fig. 2**
>
> All the figures have been fixed and improved.
>
> > **While I am happy the paper comes with code, for the sake of reproducibility I would urge the authors to structure their code [...]**
>
> That's a great feedback! We'll surely try to improve our code and make it more readable and reproducible! This is very important for the ML community as a whole.

---

### Author Response · Authors · 2023-08-17
**Summary of the Major Changes to the Paper (definitions, figures, ...etc)**

We thank all the reviewers and the action editor for the detailed feedback and valuable comments! We are glad that reviewers found our work to be: **very well written**, **interesting** with a **reasonable mix of datasets**, having **high-quality figures**, **polished**, an **enjoyable read**, **relevant across multiple domains** in ML, **challenges current assumptions** in the field, an **important contribution**, motivating the **important habit of checking for spurious features**!

The revised manuscript has been uploaded, and the changes in the text are highlighted. We will also provide individual responses to each of the reviewers. We address common concerns below.

### **1. Precise Definition of Shortcut and Avoiding Circular Claims**

We are sorry for the confusion caused by the definition of "shortcut". This term has been widely used in the literature, but in slightly different ways, and there is lack of a clear defintion. To simplify our presentation and avoid confusion, we have removed the term "shortcut," and we express our ideas simply in terms of spurious features.

*The logical thread below will clarify that our work has no circular claims.*

**(a) Train/Test Data:** For any ML task, we first need to define the training data (data used for training the model) and test data (data on which the model will be deployed/tested).

**(b) Spurious Feature:** A latent feature $s$ is spurious if it is correlated with the label $y$ in the training data but not in the test data. (Notice that changing the train or test data may affect whether a given feature is spurious or not.)

**(c) Generalization:** A model that performs well on both train and test data is said to have a good generalization. Whereas, a model that performs well on training data but poorly on the test data is said to have a poor generalization.

**(d) Types of Spurious Features:** The common conception is that spurious features hurt generalization. However, we show that not all spurious features hurt generalization. Given a task, model $\mathcal{M}$, and dataset $D$, the spurious feature $s$ can be of two types: *"benign"* (does not hurt model generalization) or *"harmful"* (hurts model generalization). For a given classification task, a spurious feature $s$ in dataset $D$ can be benign (harmful) for $\mathcal{M}$ depending upon whether the spurious feature is *"harder"* (*"easier"*) to learn than the core features.

**(e) Instance Difficulty Metrics:** To quantify/measure "easiness", we use instance difficulty metrics like Prediction Depth (PD) or $\mathcal{V}$-Usable Information. What is easy for one model may be hard for another (see Fig-4). So the definition of "easy" is highly intertwined with the model under consideration. We, therefore, use metrics like PD or Usable-information, which take the model architecture into account and account for this dependence on the model.

**(f) Proof of Concept:** Please refer to Fig-5 and Table-1 in experiment section-4.1 (Not all spurious features hurt generalization!). This experiment illustrates very clearly that spurious features that are harder (as measured using PD) than core features are "benign" in nature, as the model does not learn them. Such features do not hurt model generalization and maintain a high test accuracy. It is also clear from the results that spurious features that are easier than core features hurt model generalization and result in poor test accuracy.

**(g) Detecting Spurious Features:** We use the above observations and empirical evidence to develop a novel method to detect spurious features during training. Using instance-difficulty metrics like PD, we monitor easy features learned by the initial layers of a DNN early during training to identify harmful spurious features.

Please refer to Sec-3 (Background and Methodology) for a more mathematical formulation of the above definitions.

### **2. Figures and Font Size**

We have improved the readability, clarity, and font sizes in all the figures.

### **3. PD Computation and Equation**

We have added Fig-2 to illustrate how PD is computed. The Eq-1 for PD has also been updated/modified to make it more clear and easier to interpret.

$\text{PD}(x) = \min_k [ k | f_{knn}^k(x) = f_{knn}^i(x); i>k ] $

$f_{knn}$ is the $k$-NN classifier, $\phi^i$ is the feature embedding for the given input at layer-$i$, and $N$ is the index of the final layer of the model. The PD of an input is defined as the minimum number of layers the model requires to classify the input. The lower the PD of input, the easier it is to classify. It is computed by building $k$-NN classifiers on the embedding layers of the model, and the earliest layer after which all subsequent $k$-NN predictions remain the same is the PD of the input.

---

### Decision · Action_Editors · 2023-09-14

**Recommendation:** Accept with minor revision

**Comment:**

Reviewers found the initial manuscript, though handling interesting problem, unclear and lacking important experiments to support the claims of the paper. Authors in discussion with the reviewers updated the manuscript including important additional experiments and rewriting the paper to make it more understandable. In words of one of the reviewer from discussion, "Given the updated manuscript with a clearer definition of the material, as well as the added analysis with the alternative, simple model-based instance difficulty and the multiple trials for the PD-plots, I feel confident that this manuscript is in a good shape and sufficiently supports its claims. It motivates the importance of checking at least for obvious spurious correlations picked up by the model, which the test accuracy does not reflect, and thus provides a sufficient contribution to the field."

I am happy to recommend acceptance of the current draft with minor revision to add more references as mentioned during the discussion phase.

**Audience:**

Feature learning and spurious features are an important topic in training of deep neural networks.

**Claims And Evidence:**

The updated manuscript has experiments to support the claims of the paper.

---

> ### Author Response · Authors · 2023-09-18
> **Thank you so much!**
>
> Dear Action Editors and Reviewers,
>
> We thank you very much for the decision and are happy that our paper was accepted.
>
> We will surely make the requested changes you've asked for and submit the camera-ready version.
>
> Thank you!